# Catchment scale assessment of drought impact on environmental flow in the Indus Basin, Pakistan

Khalil Ur Rahman[1,2], Songhao Shang[2], Khaled S. Balkhair[3], Hamza Farooq Gabriel[4], Khan Zaib Jadoon[5], Kifayat Zaman[6,7]

[1]Department of Hydraulic Engineering, School of Civil Engineering, Shandong University, Jinan, Shandong, 250061, China

[2]State Key Laboratory of Hydroscience and Engineering, Department of Hydraulic Engineering, Tsinghua University, Beijing 100084, China

[3]Department of Hydrology and Water Resources Management, King Abdulaziz University, P.O Box 80208, Jeddah 21589, Saudi Arabia

[4]School of Civil and Environmental Engineering, National University of Sciences and Technology, Islamabad 44000, Pakistan

[5]Department of Civil Engineering, Islamic International University, Islamabad 44000, Pakistan

[6]Federal Water Management Cell, Ministry of National food security and research, Islamabad 44000, Pakistan

[7]Department of Soil and Climate Sciences, University of Haripur, 22660, Khyber Pakhtunkhwa, Pakistan

*Correspondence to*: Songhao Shang (shangsh@tsinghua.edu.cn)

**Abstract.** The impact of drought on environmental flow (EF) in 27 catchments of the Indus basin is studied from 1980-2018 using the Indicators of Hydrologic Alterations (IHA). Standardized Precipitation Evapotranspiration Index (SPEI) was systematically propagated from one catchment to another using principal component analysis (PCA). Threshold regression is used to determine the severity of drought (scenario-1, drought severity that causes low flows) and month (scenario-2, months where drought has resulted in low flows) that trigger low flows in the Indus Basin. The impact of drought on low EFs is quantified using Range of variability analysis (RVA), which is an integrated component of IHA used to study the hydrological alterations in environmental flow components (EFCs) by comparing the pre- and post-impact periods of human and/or climate interventions in EFCs. Hydrological alteration factor (HAF) is calculated for each catchment in the Indus basin. The results show that most of the catchments are vulnerable to drought during the periods 1984-1986, 1991/1992, 1997 to 2003, 2007 to 2008, 2012 to 2013, and 2017 to 2018. On a longer time scale (SPEI-12), drought is more severe in Lower Indus Basin (LIB) than the Upper Indus Basin (UIB). IHA pointed out that drought significantly impacts the distribution of environmental flow components, particularly extreme low flow (ELF) and low flow (LF). The magnitude and frequency of the ELF and LF events increase as drought severity increases. The threshold regression provided useful insights indicating that moderate drought can trigger ELF and LF at shorter time scales (SPEI-1 and SPEI-6) in the UIB and Middle Indus Basin (MIB). Conversely, severe and extreme drought triggers ELF and LF at longer time scales (SPEI-12) in LIB. The threshold regression also divided the entire study period (1980-2018) into different time periods (scenario-2), which is useful in quantifying the impact of drought on low EFs using the SPEI coefficient. Higher SPEI coefficients are observed in LIB, indicating high alterations in EF due to

drought. HAF showed high alterations in EF in most of the catchments throughout the year except in August and September. Overall, this study provided useful insights for analyzing the effects of drought on EF, especially during low flows.

## 1 Introduction

Environmental flow (EF) refers to the quantity, timing and quality of freshwater flows in rivers that are necessary to support/sustain ecosystem services, e.g., aquatic life, human requirements, biodiversity, and livelihoods, etc. (Arthington et al., 2018; Virkki et al., 2022). However, EFs are under moderate to severe threat due to the rapidly growing population, anthropogenic activities (i.e., damming and flow regulations), and climate and land use changes (Benjankar et al., 2018; Best, 2019; Gudmundsson et al., 2021; Pardo-Loaiza et al., 2022). On a global scale, it is estimated that approximately 65% of the

discharge (in terms of quantity) in rivers poses a moderate to a severe threat to biodiversity (Vörösmarty et al., 2010), connectivity of 48% of rivers is diminished (Grill et al., 2019), and fish biodiversity has been significantly altered in 53% of the rivers (Su et al., 2021). The main causes of such degradation and alteration in river flow regimes around the globe are associated with anthropogenic activities and climate change (Richter et al., 2006; Stamou et al., 2018; Wineland et al., 2021). Therefore, there is a need to re-think and properly manage the water resources in regions subjected to water scarcity and, most

importantly, severe changes in regional climate.

The Indus River basin is one of the typical and most depleted basins due to substantial climate and land use changes, resulting in limited water availability (Azmat et al., 2019; Immerzeel et al., 2010; Laghari et al., 2012; Sharma et al., 2010). Upper Indus Basin (UIB) is the hotspot for climate change, whereas Middle Indus Basin (MIB) and Lower Indus Basin (LIB) are dependent on the availability of water from UIB. Several studies have reported an increase in the future precipitation and temperature

(Forsythe et al., 2014; Nepal and Shrestha 2015; Rajbhandari et al., 2015); however, Shahid and Rahman (2021) reported that the findings in most of the studies are not consistent with global trends due to a number of reasons. Precipitation in the Indus Basin is highly erratic and decreasing over time (Rahman et al., 2020a), while temperature has shown an increasing trend, which consequently resulted in a decreased river flow over time (Dahri et al., 2021; Shahid and Rahman, 2021). The erratic nature of precipitation and increased temperature resulted in a significant decline in riverine flows (i.e., 90% reduction in flow

to the Indus Delta)  due to the hydrological alterations in flow regime (Salik et al., 2016; Syvitski et al., 2013). Therefore, the limited availability of surface water has substantially increased groundwater withdrawal (Rahman et al., 2022a), which poses severe threats to sustainable surface and groundwater management in the Indus Basin. In conclusion, freshwater resources are highly vulnerable to climate and land use changes in the Indus Basin, where EF can serve as an integral component for sustainable water management.

EFs in the Indus Basin can be severely impacted by climate change through shifts in precipitation (pattern and intensity), temperature, glaciers melting, and extreme weather events (Immerzeel et al., 2015; Rees and Collins, 2006). Pakistan (i.e., Indus Basin) is highly vulnerable to climate change and placed at 8[th] position among the countries most affected by climate change (Eckstein et al., 2018). Therefore, the Indus Basin experienced more frequent and severe extreme events in the recent

few decades. Among these extreme events, drought is the major one and is experienced most frequently (three per decade) due to its arid and hyper-arid nature (Ahmed et al., 2020). Drought is broadly classified into four major classes, including meteorological, hydrological, agricultural, and socio-economic droughts (Stephan et al., 2021). Several studies reported that the intensity of drought increases from UIB to LIB, where the climate (temperature) plays an important role (Rahman et al., 2022b). Similar to meteorological drought, the severity and duration of hydrological drought are higher in LIB compared with UIB (Rahman et al., 2022b). The persistent meteorological drought results in a hydrological drought, resulting in a decrease in water availability and, thus, insufficient EFs (Peña-Guerrero et al., 2020). This implies that drought can alter the distribution of EFs both spatially and temporally to whom the Indus Basin will be extremely vulnerable, particularly LIB in arid and hyper-arid areas.

The intensity and frequency of droughts are increasing around the world and particularly in the Indus Basin (Chiang et al., 2021; Vicente-Serrano et al., 2019; Wen et al., 2019); therefore, it is extremely important to analyze the impact of drought on water availability, especially the variations and alterations in EFs. Very few studies have assessed the impact of drought (meteorological) on EF. For instance, Młyński et al. (2021) have studied the impact of drought (Standardized Precipitation Index, SPI) on EF across mountainous catchments in Poland. The study reported that drought has the potential to alter the EF, whereas the alterations in EF are dependent on several factors such as topography (slope), local climate, and hydrogeological conditions. However, the impact of drought on EFs is yet to be investigated in details. To the best of our knowledge, no such study quantified the alterations in river flow due to drought and identified thresholds (drought severity and month) that can trigger the alterations in river flow and result in low EFs. Bearing in mind the importance of conserving minimum flow in rivers to protect the ecosystem, this study for the first time evaluates the impact of drought on EF using the Indicators of Hydrologic Alterations (IHA).

The objectives of the current study are (i) assessing the environmental flow components (EFC), particularly extreme low flow and low flow, for the 27 catchments of the Indus Basin, (ii) investigating the drought severity and drought months that trigger low EFs in the Indus River using threshold regression, (iii) application of the range of variability analysis (RVA) to quantify the impact of drought on low EFs, and (iv) analyzing the degree of alterations in each catchment using the hydrological alteration factor (HAF).

## 2 Study area

Indus Basin is the 12[th] largest basin in the world and is situated in four countries, including Pakistan, China, India, and Afghanistan (Laghari et al., 2012). The largest part of Indus Basin lies in Pakistan, covering an area of 855,045 km$^2$ between 66.20°–82.50°E and 24.02°–37.07°N. Indus Basin in Pakistan has a complex topography and diverse climate, where more than 40% of the Indus Basin has an elevation greater than 2,000 m (Rahman et al., 2022b). Based on climate and topography, Indus Basin is classified into UIB, Middle Indus Basin (MIB), and LIB (shown in Fig. 1) following the demarcation from Aftab et al., (2022), Rajbhandari et al., (2015), and Shahid et al., (2021).

UIB is the glacial region of the Indus Basin having arid climatic nature and comprised of permanent snow and glacier reservoirs. UIB is comprised of the famous Hindu-Kush-Himalayas Mountain ranges, which are the origin of freshwater in the Indus River and its tributaries (Laghari et al., 2012; Rahman et al., 2022b).

MIB has a humid to arid climate comprising of Indus Plain, and most of the MIB area consists of a well-developed irrigation network. The entire Indus Basin has 228,694 km$^2$ (21% of the basin area) of irrigated area, where 60.9% is situated in Pakistan (Laghari et al., 2012). The Indus Basin Irrigation System (IBIS), one of the largest irrigation networks in the world, covers most of the area in MIB (Rahman et al., 2022b). IBIS is one of the integral parts of sustainable water and food supply in Pakistan because it supports approximately 90% of Pakistan's agricultural production (Yang et al., 2013). LIB is located downstream of the Indus Basin, which covers the Indus Plain and Indus Delta, and the climate varies from arid to hyper-arid (Young et al., 2019). Indus Plain in MIB and LIB is covered by the Indus River and several other major rivers in the west, including Sutlej, Jhelum, Chenab, and Ravi (Kalair et al., 2019).

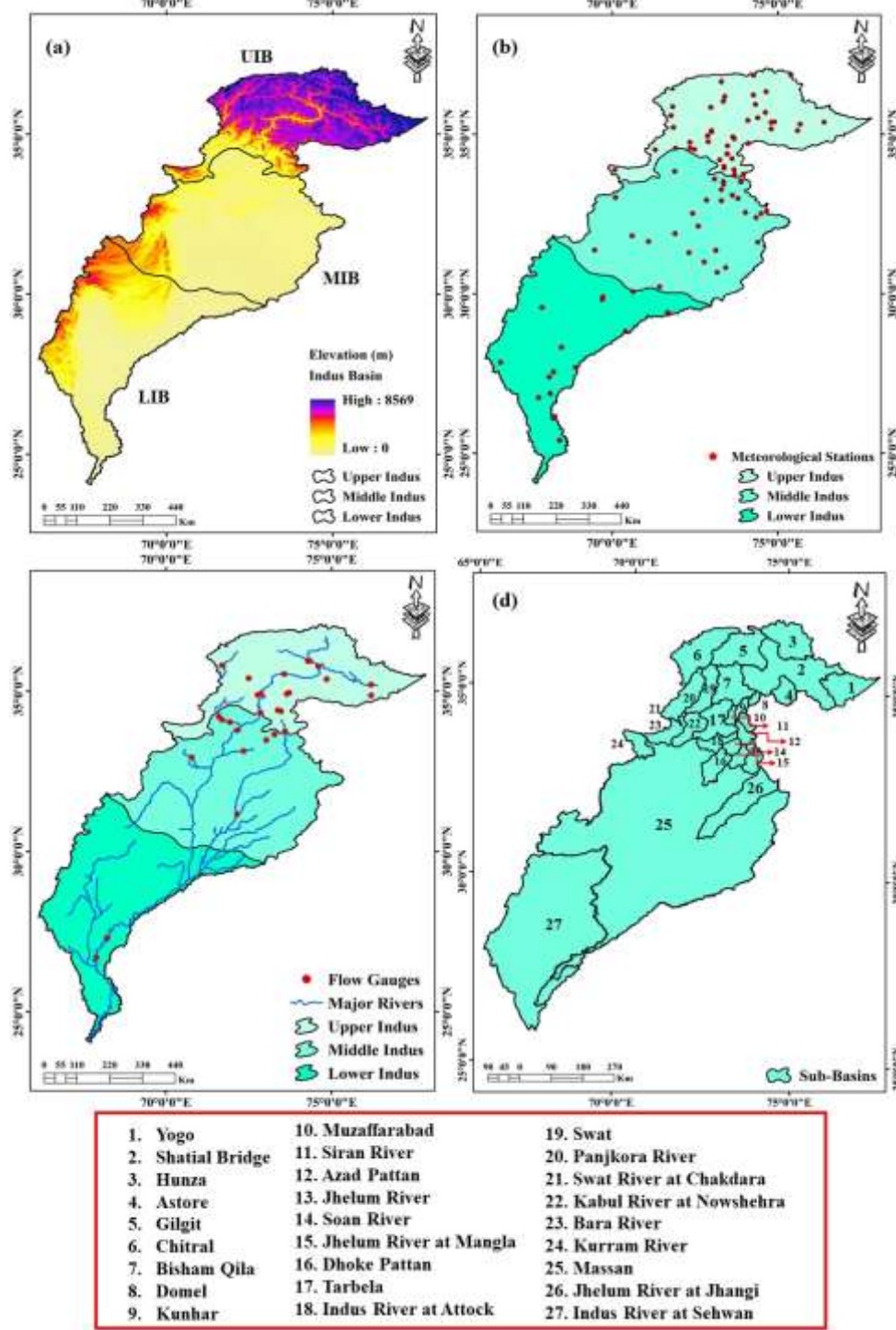

| | | |
|---|---|---|
| 1. Yogo | 10. Muzaffarabad | 19. Swat |
| 2. Shatial Bridge | 11. Siran River | 20. Panjkora River |
| 3. Hunza | 12. Azad Pattan | 21. Swat River at Chakdara |
| 4. Astore | 13. Jhelum River | 22. Kabul River at Nowshehra |
| 5. Gilgit | 14. Soan River | 23. Bara River |
| 6. Chitral | 15. Jhelum River at Mangla | 24. Kurram River |
| 7. Bisham Qila | 16. Dhoke Pattan | 25. Massan |
| 8. Domel | 17. Tarbela | 26. Jhelum River at Jhangi |
| 9. Kunhar | 18. Indus River at Attock | 27. Indus River at Sehwan |

**Figure 1.** (a) Division of Indus basin into UIB, MIB, and LIB with elevation (m), (b) distribution of rain gauges (RGs) and temperature stations, (c) distribution of flow stations, and (d) delineated catchments of the Indus basin

UIB is characterized by mild precipitation, low temperature, and thus low potential evapotranspiration (PET). UIB (areas between 34–36°N) receives less than 100 mm of precipitation during the monsoon season (Rahman et al., 2020a), while the downstream (southern UIB) receive relatively higher precipitation. On the other hand, MIB has humid climatic nature and receives more than 700 mm of precipitation during the monsoon season. The precipitation decreases to less than 100 mm from MIB to LIB, especially between 24 to 28°N (Iqbal and Athar, 2018). The temperature in LIB and southern MIB is getting warmer, making these regions more vulnerable to severe and frequent drought events (Rahman et al., 2022b). Overall, Indus Basin receives maximum precipitation of approximately 1500 mm/a in the mountainous regions while less precipitation of about 100 mm/a in the Indus Plain (Dimri et al., 2015). The high temperature and low precipitation make the Indus Basin, especially the LIB, heavily dependent on freshwater availability from UIB (Laghari et al., 2012).

Major rivers of Pakistan, including the transboundary rivers such as the Kabul River, Jhelum, Ravi, Sutlej, and Chenab, contribute approximately 70% of freshwater to the Indus Basin (Karimi et al., 2013; Young et al., 2019). The above-mentioned rivers along with the Indus River serves as a source of water for irrigation and are extremely critical for LIB (Masood et al., 2020). However, river flow in the Indus Basin is highly seasonal depending upon the temperature and precipitation intensity, i.e., low flow in winter and high flow in summer due to glacial melt (Ali et al., 2009). Extreme events induced by climate change, such as drought, has a substantial impact on the river flows where most of the studies reported a decreasing trend in river flow in different parts of the Indus Basin (Azmat et al., 2020; Hasson et al., 2017; Mukhopadhyay et al., 2015; Shahid and Rahman, 2021; Shrestha et al., 2019).

## 3 Datasets and Methodology

The schematic diagram of methods used in the current study is shown in Fig. 2. The methodology is broadly divided into two main categories, i.e., estimation of environmental flow components (EFCs) and assessing the impact of drought on environmental flow. IHA consists a total of 67 parameters, which are grouped into IHA parameters (33) and EFC (34). The EFC parameters are grouped into five main EFC classes, i) extreme low flow (ELF), ii) low flow (LF), iii) high flow pulses, iv) small floods, and v) large floods (https://www.conservationgateway.org/Documents/IHAV7.pdf). Out of the five EFC classes, the first two classes are of our concern because the flow in rivers are minimum during the drought period and they may threaten the survival of biodiversity and harm the ecosystem when the river flow reduces. On the other hand, drought is estimated using SPEI and systematically propagated from one catchment to a downstream one using PCA. Drought is assessed at three-time scales, i.e., short-term (1 month) using SPEI-1, seasonal (6 months) using SPEI-6, and long-term (12 months) using SPEI-12. The impact of drought on ELF and LF is assessed using threshold regression. Threshold regression is used to identify the drought severity that triggers ELF and LF at the catchment scale. Moreover, the months of ELF and LF under the influence of drought are also assessed using threshold regression. Finally, RVA analyses are used to appraise the impact of drought on environmental flow in each catchment of the Indus Basin.

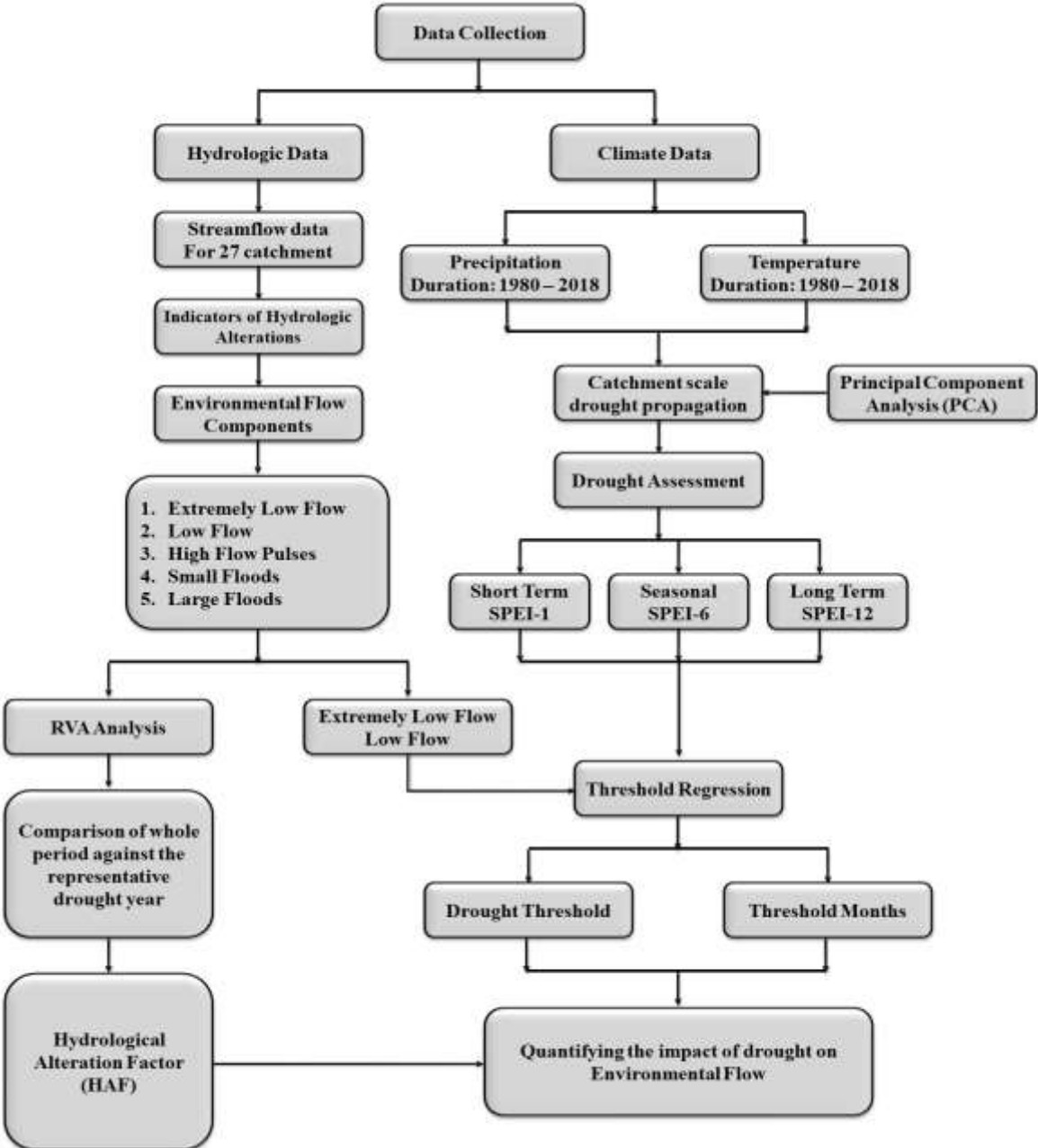

**Figure 2.** Methodological framework adopted in the current study

### 3.1 Datasets

The temperature and precipitation data used to calculate drought (SPEI) at 79 climate stations and rain gauges (RGs) (Fig. 1)
was acquired from the Pakistan Meteorology Department (PMD) and Water and Power Development Authority (WAPDA). A

high proportion of data was acquired from PMD, i.e., 61 stations/RGs, while the remaining 18 stations were from WAPDA. Stations/RGs collected from WAPDA are operated under the Snow and Ice Hydrology Project (SIHP) and mostly located in UIB and in the elevated regions of MIB (Rahman et al. 2022a). The river flow data at 27 flow stations are collected solely from WAPDA. After thoroughly analyzing all the collected data (i.e., checking the date/years of available data at most of the in-situ stations), a period from 1980–2018 is chosen to demonstrate the drought impact on environmental flow where all the in-situ stations have the data with few or no missing values. However, few catchments have the data for shorter period of time, e.g., Indus River at Shatial Bridge (1984–2014), Hunza catchment (1995–2018), and Indus River at Tarbela (1983–2015). Detailed information about the data collected is given in Table 1.

**Table 1.** Detailed information about the data collected

| No. | Data | Sub-basin | Duration | Authority |
|---|---|---|---|---|
| 1 | Precipitation | UIB/MIB | 1980–2018 | PMD/WAPDA |
|   |   | LIB | 1980–2018 | PMD |
| 2 | Temperature | UIB/MIB | 1980–2018 | PMD/WAPDA |
|   |   | LIB | 1980–2018 | PMD |
| 3 | River Flow | UIB/MIB/LIB | 1980-2018 | WAPDA |

Data in Pakistan (Indus Basin) is usually manually collected by PMD and WAPDA. Therefore, the collected data has several issues, including errors due to personal and instrumental errors, splashing due to climate, errors due to winds, topography, etc. These errors result in poor quality and missing data. The initial attempts are made by PMD and WAPDA to rectify the data following the standard code of WMO-N issued by the World Meteorological Organization. Besides, we have also performed data quality tests including the kurtosis and skewness methods to check the data quality (Tables S1 and S2), and the missing data is filled by zero-order methods following Rahman et al. (2020a).

**3.2 Estimation and propagation of drought**

Indus Basin of Pakistan has a data scarcity issue, where RGs/stations are sparsely distributed and not enough to represent the local climate. Therefore, PCA is used to calculate the principal components of precipitation and temperature before the estimation of drought. In this study, we followed the procedure recommended by Rahman et al. (2023a) to systematically propagate drought from one catchment to another, i.e., from catchment 1 (Yugo) to catchment 27 (the Indus River at Sehwan catchment). However, it was ensured that the maximum variance is retained in the principal components estimated from RGs/stations inside the particular catchment. This step helped us to retain the maximum information about the catchment while including the influence of surrounding catchments. Overall, the computed representative datasets (principal components) of precipitation and temperature have a linear combination that reflects original RGs/station data information.

Drought in this study is appraised using the most widely used SPEI index (Vicente-Serrano et al., 2010), which is developed using the Standardized Precipitation Index (SPI) algorithm proposed by McKee et al. (1993). The principal components of

precipitation and temperature propagated from upstream to downstream of the Indus Basin are used to compute SPEI. Most of the studies recommended the application of SPEI because it uses both temperature and precipitation data to calculate water balance and estimate the surplus water (Liang et al., 2021; Liu et al., 2019; Rahman et al., 2022b). Furthermore, SPEI also considers the variations in climate by avoiding too many zeros in precipitation estimates that are true particularly across arid and hyper-arid regions (Wu and Qian, 2017), especially across the Indus Basin. Besides, SPEI has better distribution fitting and thus better capture the drought severity (Stagge et al., 2015). Following Rahman et al. (2022b), log-logistic distribution is used to compute SPEI to better reflect drought at the catchment scale.

SPEI in this study is estimated at different time-scales, i.e., SPEI-1, SPEI-6, and SPEI-12 representing short-term (1 month), seasonal (6 months), and long-term (12 months) drought events, respectively. The time period is selected based on the climatological and hydrological characteristics of the Indus Basin, as the river flows in UIB and MIB are extremely seasonal and subjected to significant hydrological alterations (dam operation and water diversion to IBIS). The severity of SPEI generally ranges from -2 to 2, where the drought and wet events are represented by negative and positive SPEI values, respectively. However, this study uses a threshold value of SPEI<-1.0 to differentiate the drought-impact period for RVA analyses. When the SPEI value is in the range of -1.0 to -1.5, -1.5 to -2.0, or less than -2.0, the drought event is classified as extreme drought, severe drought, or moderate drought (Table S3), respectively.

### 3.3 Indicators of Hydrologic Alterations (IHA)

Nature Conservancy has developed the IHA (http://www.nature.org/), which has been successfully used to quantify the alterations in river flows (Lee et al., 2014; Nature Conservancy, 2007; Rahman et al., 2020b; Richter et al., 1996). Assessing the hydrological alterations in river flows is extremely important for sustainable water resource management, quantifying anthropogenic impacts on river flow and associated ecology, and maintaining a healthy ecosystem (Hart and Breaker, 2019; Lytle and Poff, 2004; Poff and Zimmerman, 2010). IHA is gaining more attention nowadays and has been used in several hydrological applications, including ecology, water resources management, assessing alterations in streamflow, and others (Lee et al., 2014; Mathews and Richter, 2007; Rahman et al., 2020b).

IHA consists of a total of 67 parameters, categorized into two groups, i.e., hydrologic (33 parameters) and EFC (34 parameters). IHA characterizes the inter- and intra-annual variations in river flows based on 33 hydrologic parameters following the five major flow regimes; i) the magnitude of monthly flows, ii) duration and magnitude of annual extreme flows, iii) timing of extreme flows, iv) duration and frequency of low and high flow pulses, and v) frequency and magnitude of changes in flow (Mathews and Richter, 2007). IHA categorizes streamflow into several components, including low flows (where the streamflow values are less than or equal to the 25th percentile), moderate flows (where the streamflow values range between the 25th to 75th percentiles) and high flows (where the streamflow values are greater than the 75th percentile). Besides, when the flow is less than the 10th percentile, we classified it as extreme low flow. The hydrologic parameters of IHA are interconnected, i.e., these are proposed based on ecological relevance between them and these parameters reflect human-induced alterations in river flows (Arthington et al., 2006; Olden and Poff, 2003). These alterations include dam operations,

groundwater withdrawal, water diversions, and land use changes (Mathews and Richter, 2007). Further details about IHA and its parameters can be found in references (Gao et al., 2009; Nature Conservancy, 2007; Richter et al., 1996). IHA in this study is used to compute the EFC, particularly ELF and LF components in 27 catchments of the Indus Basin.

IHA is calibrated using the advanced calibration option following the guidelines mentioned in the user manual. To calibrate the IHA, it is first ensured that IHA provides a clear distinction between low flows (during the drought years) and high flows (major floods) by adjusting the EFC parameters. Since we are interested in assessing individual events (both high flows and low flows), the high and low flow thresholds were adjusted for individual flow peaks. Therefore, during the calibration process, IHA hydrographs were compared with major flood events across each catchment. After splitting the river flow into the high flow and low flow peaks, the hydrograph is further calibrated for five major EFC classes by adjusting the small and large flood minimum peaks and extreme low flow thresholds.

## 3.4 Range of Variability Approach (RVA)

Several methods have been proposed to assess the alterations in flow regimes. Among these methods, the RVA approach developed by Richter et al. (2003) and Richter et al. (1996) has been widely used to assess hydrological alterations (Pal and Sarda, 2021; Rahman et al., 2020b; Shiau et al., 2006; Zheng et al., 2021). RVA is incorporated into IHA software and is used when no or minimal ecological information is available to support the environmental flow. RVA is used to develop the initial flow management goals for river flows, illustrating the linkage between river flow and ecosystem that would accrue over a certain time and flow targets (Richter et al., 1997; Richter et al., 2003). RVA is generally used to compare the pre-impact and post-impact periods to analyze the human-induced impact on river flow regimes (hydrologic alterations).

Major steps in implementing RVA include; i) characterization of the natural range of variability in hydrologic conditions, such as rate, magnitude, frequency, and duration, ii) quantifying the degree of alterations, iii) developing the hypothesis about the impact assessment, iv) addressing the identified alterations based on proposed hypothesis, and v) implementing the designed ecosystem measures (Mathews and Richter, 2007). The hypothesis developed in this study is that drought significantly impacts the ELF and LF classes of EFCs. To investigate the impact of drought on environmental flow in current study, the whole period (1980-2018) is considered a pre-impact period (without differentiating between drought and wet events, which can also be considered as normal flow years without focusing on specific drought years), while the specific drought years (i.e., years with the average SPEI values of less than -1, also considered as representative drought years as identified by Rahman et al., 2023a and 2023b) are considered a post-impact period.

Hydrological alteration factor (HAF) is calculated based on the results of RVA analyses, i.e., comparing the whole period with drought years. HAF is used to demonstrate the vulnerability of environmental flow to drought in all the catchments of the Indus Basin. HAF range is divided into three main categories following Richter et al. (1997), including no alterations (0.00 < HAF < 0.33), moderate alterations (0.34 < HAF < 0.67), and high alterations (0.68 < HAF < 1.00). HAF is calculated using the following equation:

$$\text{HAF} = \frac{\text{Observed Frequency} - \text{Expected Frequency}}{\text{Expected Frequency}} \qquad (1)$$

where observed frequency represents the years where a particular EFC falls in a specified range, e.g., between 25th and 75th percentiles, during the drought years. The expected frequency is calculated as follows:

$$\text{Expected Frequency} = P \times N_P \qquad (2)$$

where $P$ is the probability of the specified range of EFC, i.e., 50% for the range between the 25th and 75th percentiles, and $N_p$ represents the number of drought years.

### 3.5 Threshold Regression

Threshold regression is a regression model that links the predictors with outcomes based on a threshold parameter, also known as a change point. Threshold regression provides a very interpretable and elegant way to model the non-linear relationship between the predictor and outcome (Hansen, 2011). The results from threshold regression are dependent on the threshold parameter, i.e., threshold regression can take different forms depending on the threshold parameter. Threshold regression differs from change-point analysis (Hansen, 2000; Yu, 2012), which is mostly applied to time series data and mainly detects the structural changes along the natural axis, e.g., time or space. There are several main reasons for selecting threshold regression over change-point analysis in this study. First, the threshold regression is capable to understand the non-linear relationship between the threshold variables (drought and environmental flow in our case), while the change-point analysis can be used to see the changing trend in a time-series data (for instance, we can only see the change point in drought or in environmental flow) (Hansen, 2011). In change point analyses, time series data are divided into successive sub-periods, where the relationship between outcome and predictors changes from one sub-period to another (Muggeo, 2008). Second, the threshold regression is more robust than the change-point analysis in dealing with non-linear relationship between the variables, and comparable with other non-linear regression models (e.g., spline regression model). Third, the threshold regression has the potential to adapt any shape (explained by Fong et al., 2017) depending on the threshold variable and its threshold value. Further detail about threshold regression can be found in Hansen (2011).

In this study, threshold regression is applied to study two different scenarios; 1) to determine the drought severity (classified into different classes following the recommendations from McKee et al. (1993)) that causes ELF and LF in different catchments of the Indus Basin, and 2) to determine the months where drought has caused the ELF and LF in Indus Basin. Two different threshold parameters are considered to achieve the above two goals, i.e., drought severity (SPEI) and month (time).

$$y_t = x_t \beta + z_t \delta + \varepsilon_t \qquad (3)$$

where $y_t$ is dependent variable (EFC), $x_t$ is a vector of independent variables (time/month for scenario-1 and SPEI for scenario-2), $z_t$ is threshold variable (SPEI for scenario-1 and time/month for scenario-2), $\varepsilon_t$ is independent and identically distributed (IID) error with mean 0 and variance $\sigma^2$, and $\beta$ and $\delta$ are the coefficients of the corresponding variables.

## 4 Results

Following the methodology shown in Fig. 2, the results section is mainly divided into time series assessment of drought, distribution of EFC in selected catchments of the Indus Basin, quantifying the drought impact on EFC (i.e., ELF and LF), and RVA analysis to investigate the drought impact on ELF and LF (alterations in river flow at catchment scale).

### 4.1 Evaluation of drought in representative catchments of the Indus Basin

The temporal variations of drought at short-term (SPEI-1), seasonal (SPEI-6), and long-term (SPEI-12) time scales in representative catchments of the Indus Basin are shown in Figs. 3–5, respectively. The selected representative catchments are Gilgt, Hunza, Indus River at Bisham Qila and Shatial Bridge in UIB, Indus River at Tarbela (outflow), Indus River at Attock, Jhelum River, and Kabul River at Nowshehra in MIB, and, Indus River at Sehwan in LIB. The temporal variations in SPEI-1 (Fig. 3) show that catchments in the Indus Basin were vulnerable to drought in 1986, 1991, 1997–2003, 2007–2008, 2012–2013, and 2017–2018. However, no consistent drought trend is observed in SPEI-1 because of its relatively short duration. The number of extreme, severe, and moderate drought events in UIB are 11, 54, and 202 out of 468 months. Similarly, the extreme, severe, and moderate drought events in MIB (LIB) are 5 (27), 40 (63), and 181 (199), respectively. Overall, the severity and frequency of drought events are highest in LIB, followed by UIB and MIB.

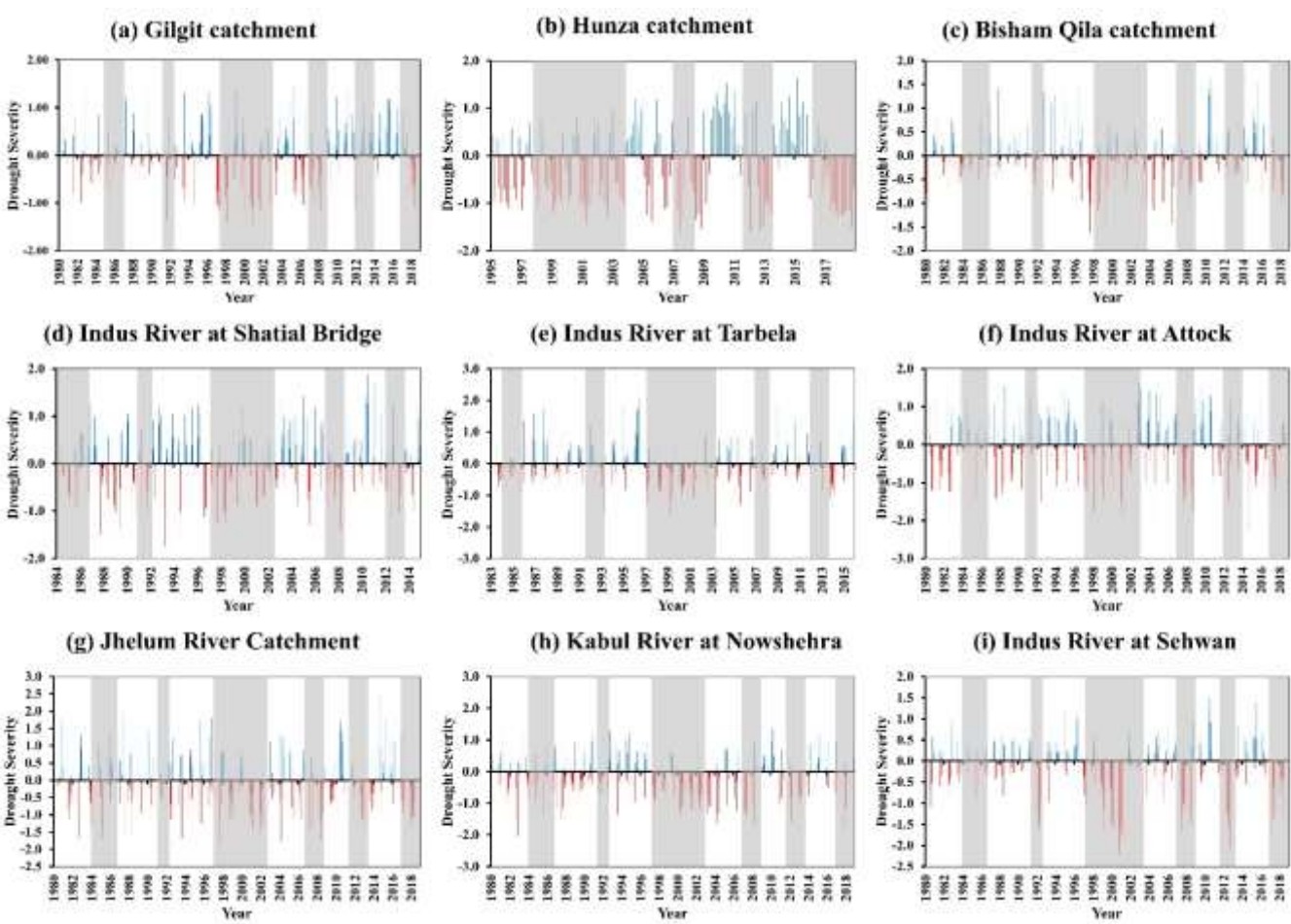

**Figure 3.** Temporal variations in SPEI-1 across the representative catchments of the Indus Basin

The temporal variation of SPEI-6 in the representative catchments of the Indus Basin is shown in Fig. 4. The vulnerable drought years at a 6-month time scale are 1984–1986, 1991/1992, 1997–2003, 2007–2008, 2012–2013, and 2017–2018 (highlighted in the shaded portion). SPEI-6 follows a similar trend to that of SPEI-1, i.e., frequency and severity of drought events are highest in LIB, followed by UIB and MIB. Drought severity is high in the Indus River at Sehwan catchment and Kabul River at Nowshehra. There is the highest number of extreme events in LIB, followed by UIB sub-basins of the Indus Basin. For instance, there are 36 (15), 98 (67), and 170 (141) events of extreme, severe, and moderate droughts in LIB (UIB), respectively. However, the number significantly decreases to 9 (extreme), 55 (severe), and 150 (moderate) in MIB.

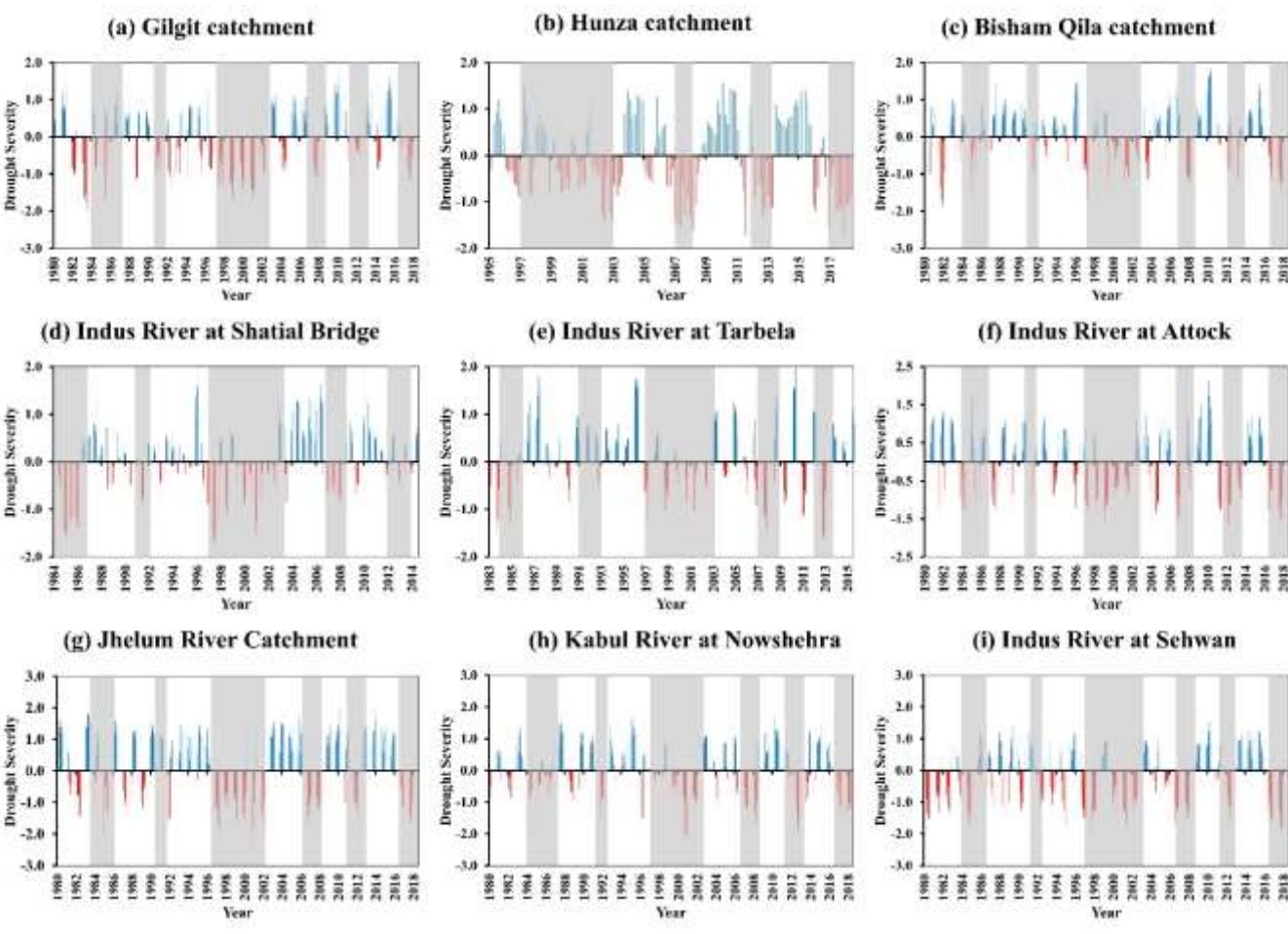

**Figure 4.** Temporal variations in SPEI-6 across the representative catchments of the Indus Basin

The drought and wet periods are more apparent on a 12-month scale than 6-month and 1-month (Fig. 5). SPEI-12 depicted the same drought period as SPEI-6, where catchments in the Indus Basin were more vulnerable to drought during 1984–1986, 1991/1992, 1997–2003, 2007–2008, 2012–2013, and 2017–2018. The figure shows that Gilgit and Indus River at Bisham Qila

catchments are more vulnerable to frequent and severe drought events compared with other catchments in UIB. The severity and frequency of drought increase from MIB to LIB, which is more evident across Kabul River at Nowshehra and Indus River at Sehwan catchments. These catchments showed high vulnerability to drought due to their arid and hyper-arid climatic nature. The average number of extreme, severe, and moderate drought events decreases from UIB (18, 77, and 144) to MIB (15, 68, and 117). However, the number of extreme, severe, and moderate drought events in LIB are 44, 104, and 172, respectively.

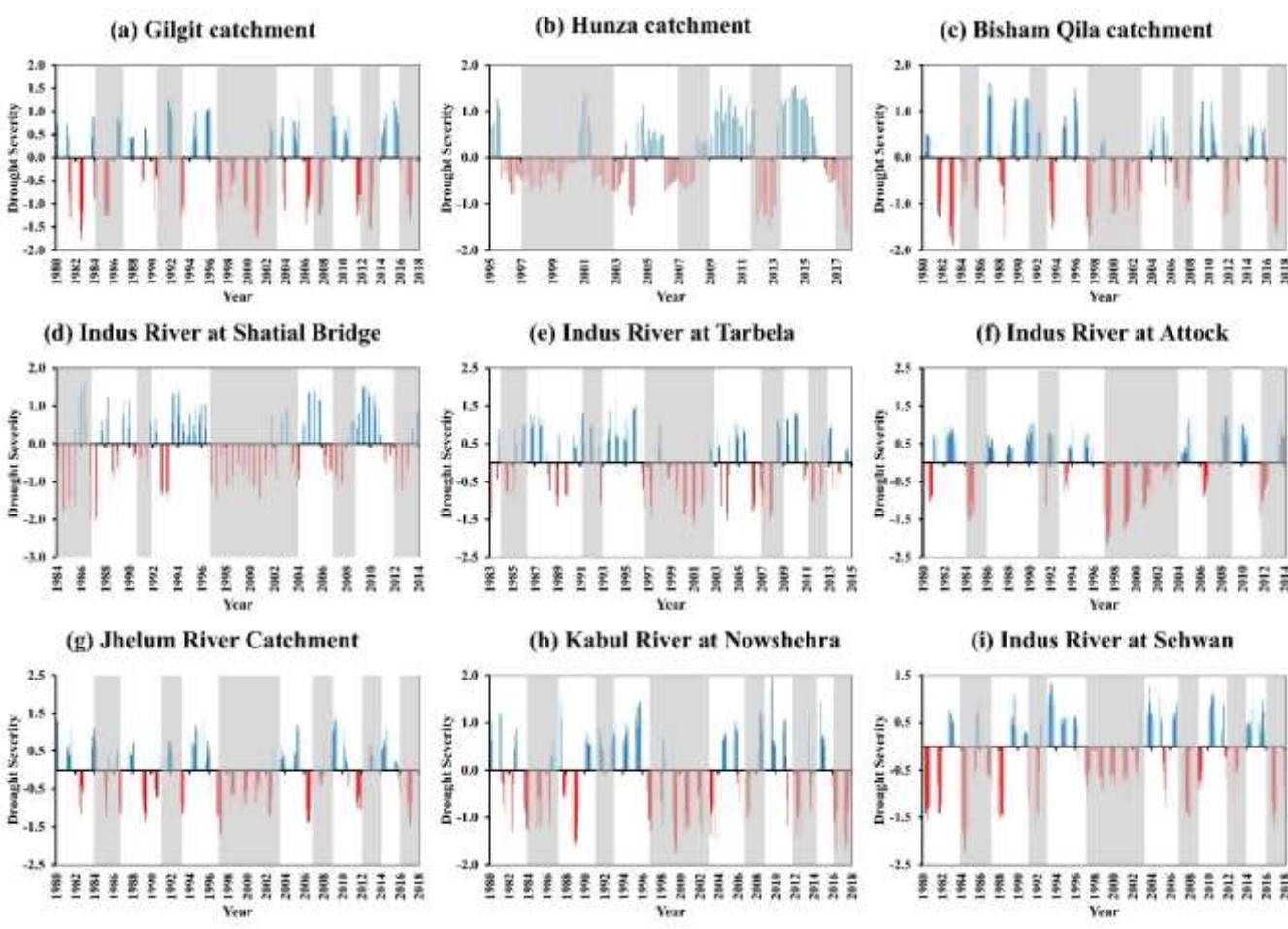

**Figure 5.** Temporal variations in SPEI-12 across the representative catchment of the Indus Basin

The variability and vulnerability of drought in each catchment are subjected to the topography and local climate of the catchment. For example, catchments in UIB are comparatively less vulnerable to extreme and severe drought than LIB because of relatively more precipitation and lower temperature. More frequent severe and extreme droughts are observed in LIB, which

is characterized by high temperature (reaches 50 °C in summer) and low precipitation (annual average below 100 mm) (Dimri et al., 2015; Rahman et al., 2022a). MIB, being the humid region, is less vulnerable to drought compared to UIB and LIB, where the precipitation is high, i.e., precipitation is more than 700 mm during monsoon season (the annual precipitation ranges from 300 mm in the south to 800 mm in north and northeast of humid region), and PET is comparatively less. However, it is worth mentioning that this study did not consider the entire hyper-arid region, and drought is propagated from UIB to LIB

using PCA; thus, the drought severity is comparatively lower. The results from this study are consistent with previous studies, including Adnan et al. (2017) and Rahman et al. (2021) that reported 1997–2003, 2007–2008, 2012–2013, and 2017–2018

being the major drought years. These studies also reported that drought is more severe in arid and hyper-arid regions compared to humid and sub-humid regions (MIB) of the Indus Basin.

## 4.2 Environmental Flow Components (EFCs) of the Indus Basin

EFCs for the representative catchments of the Indus Basin are shown in Fig. 6, where EFCs are mainly divided into ELF, LF, high flow pulses, small floods, and large floods. All the catchments show a significant reduction in the magnitude of river flow during the drought years. For instance, flow reduction is clearly visible in 1986, 1991, 1998–2002 (except for a few catchments in UIB), 2007–2008, and 2017–2018. The magnitude of ELF and LF is comparatively low in UIB, which is increasing in magnitude towards MIB (Indus River at Tarbela and Attock, and Kabul River at Nowshehra catchment) and LIB (Indus River

at Sehwan). Jhelum River catchment is located in a humid region that experienced large flood events in 2010 and 2014; therefore, the ELF and LF components of EFC are comparatively low in magnitude. On the other hand, the transboundary river catchment (Kabul River at Nowshehra) and the Indus River at Attock catchment have significant fluctuations in EFCs. Besides the transboundary river issues, climate plays a critical role in the fluctuation of EFC across the Kabul River at Nowshehra catchment. However, the Indus River at Attock catchment is located beneath the Tabela dam and depends on the

flow from Tarbela dam; thus, it shows considerable fluctuations. A high magnitude of ELFs and LFs is observed in LIB in the Indus River at Sehwan catchment. Overall, the results showed that the magnitude and frequency of ELF and LF events increase with the severity of the drought, where most of the catchments show ELF and LF during drought years, especially from 1998-2003 and 2017–2018.

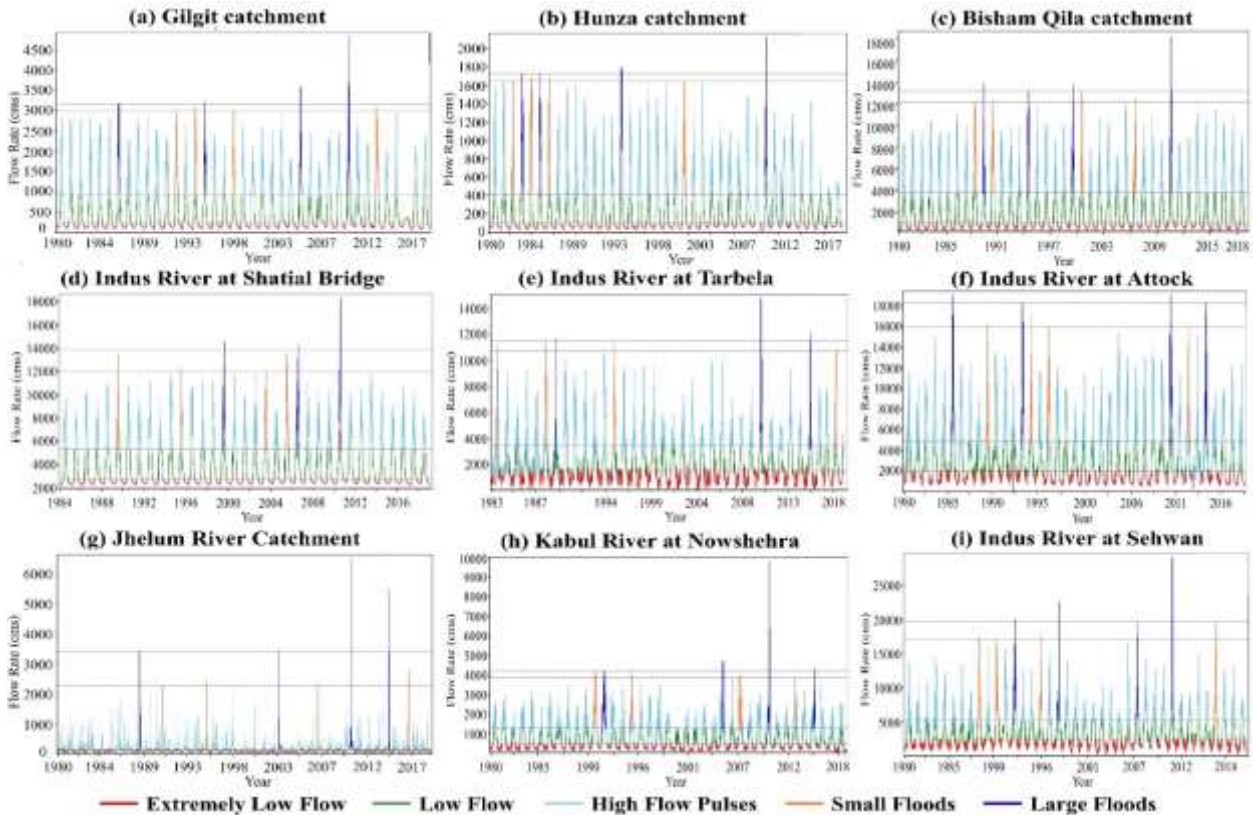

**Figure 6.** EFC components of river flow in the representative catchments of the Indus Basin

### 4.3 Assessing the impact of drought on environmental flow

Threshold regression is run under two different scenarios to quantify the impact of drought on environmental flow. The first scenario is used to determine the severity of drought that can trigger the ELF and LF events in the river flow. The second scenario illustrates the months where the drought significantly alters the environmental flow, i.e., months where consistent ELF and LF events are observed. In the first scenario, SPEI (1-, 6-, and 12-month) is considered as the threshold variable, while time (month) is considered as threshold variable in the second scenario.

#### 4.3.1 Scenario-1: Drought as a threshold variable

Table 2 shows the drought severity as a threshold for SPEI-1, SPEI-6, and SPEI-12 that causes ELF and LF events in the catchments of the Indus Basin. Most of the catchments in UIB depicted moderate drought as a threshold for SPEI-1 and SPEI-6, while a severe drought is a threshold at SPEI-12 (except for a few catchments). The results showed that the intensity of drought increases from SPEI-1 to SPEI-12 because the drought in the short term (SPEI-1) is not developed and evident (as shown in Fig. 3). In other words, frequent wet and moderate drought events are observed at short time scale. Thus, most

catchments show moderate drought as a threshold to trigger ELF and LF. However, as the time scale increases to 6 and 12 months, i.e., where precipitation is accumulated for several months, the drought becomes more evident and consistent, and

thus the severity of drought increases. Besides that, catchments in the extreme north and northeast, including Yugo, Hunza, and Astore river at Doyian catchments, demonstrated a moderate drought as a threshold to cause ELF and LF in their respective rivers irrespective of the drought severity. Indus River at Tarbela (the last catchment of UIB) depicted changes in river flow at moderate (SPEI-1) and severe (SPEI-6 and SPEI-12) drought. The threshold is relatively high for the Indus River at Tarbela and Attock catchments, which might be influenced by anthropogenic activities, e.g., the Tarbela dam operation.

Catchments in the MIB depicted relatively mild drought severity that causes changes in river flow. Most of the catchments depicted moderate drought as a threshold that triggers ELF and LF events in rivers. This is especially true for eastern catchments of the MIB (e.g., Jhelum River, Domel, Kunhar, Muzaffarabad, etc.), which have humid nature and usually drought is less as compared to western MIB, e.g., Panjkora River, Bara River, and Kurram River. Furthermore, the catchment size also contributes to lower drought severity in these catchments. The northeastern catchments (catchments from 8[th] to 16[th] shown in

Fig. 1) are subjected to land use changes, transboundary river issues, water withdrawal for IBIS and other hydraulic structures, and other anthropogenic activities (Shahid and Rahman, 2021; Siddique et al., 2018). Therefore, changes in river flow regimes across these catchments are more influenced by human-induced changes rather than climate change. Overall, the general trend in MIB is that threshold of drought severity triggering ELF and LF events increases with the time scale, i.e., from SPEI-1 to SPEI-12. Moreover, climate-induced activities also play a critical role in altering river flow regimes, e.g., particularly in Bara

River, Kurram River, Panjkora River, Swat River at Kalam and Chakdara, Kabul River at Nowshehra, Soan River, Siran River, and Jhelum River at Jhangi catchments (Rahman et al., 2022b).

**Table 2.** Threshold of drought severity that causes ELF and LF in the Indus Basin

| Catchments | Threshold | | | Catchments | Threshold | | |
|---|---|---|---|---|---|---|---|
| | SPEI-1 | SPEI-6 | SPEI-12 | | SPEI-1 | SPEI-6 | SPEI-12 |
| **UIB** | | | | **MIB** | | | |
| Gilgit | -1.162 | -1.312 | -1.621 | Jhelum River | -1.212 | -1.365 | -1.478 |
| Indus River at Bisham Qila | -1.243 | -1.375 | -1.614 | Indus River at Attock | -1.374 | -1.556 | -1.768 |
| Indus River at Tarbela | -1.305 | -1.594 | -1.887 | Kabul River at Nowshehra | -1.356 | -1.541 | -1.729 |
| Hunza | -1.293 | -1.385 | -1.544 | Domel | -1.174 | -1.356 | -1.487 |
| Indus River at Shatial Bridge | -1.176 | -1.343 | -1.605 | Muzaffarabad | -1.099 | -1.300 | -1.489 |
| Yugo | -1.157 | -1.353 | -1.497 | Azad Pattan | -1.174 | -1.330 | -1.471 |
| Astore River at Doyian | -1.204 | -1.384 | -1.478 | Jhelum River at Mangla | -1.121 | -1.325 | -1.557 |
| Chitral River | -1.215 | -1.459 | -1.739 | Kurram River | -1.428 | -1.699 | -1.836 |
| Swat River at Kalam | -1.115 | -1.478 | -1.653 | Kunhar | -1.082 | -1.297 | -1.453 |

| Swat River at Chakdara | -1.278 | -1.378 | -1.588 | Siran River | -1.398 | -1.561 | -1.772 |
|---|---|---|---|---|---|---|---|
| **Catchments** | | **Threshold** | | Soan River | -1.341 | -1.624 | -1.844 |
| | SPEI-1 | SPEI-6 | SPEI-12 | Dhoke Pattan | -1.392 | -1.581 | -1.726 |
| **LIB** | | | | Panjkora River | -1.279 | -1.525 | -1.713 |
| Indus River at Sehwan | -1.618 | -1.678 | -2.291 | Bara River | -1.240 | -1.558 | -1.737 |
| Indus River at Massan | -1.379 | -1.562 | -2.161 | Jhelum River at Jhangi | -1.147 | -1.446 | -1.648 |

Catchments in LIB are more sensitive to drought, where severe and extreme drought events are frequently observed due to a
fewer magnitude of precipitation and high temperature (Rahman et al., 2022b). Therefore, the Indus River at Massan and
Sehwan catchments depicted mostly severe and extreme drought severity as a threshold for ELF and LF in the LIB. Meanwhile,
the threshold of drought severity increases from SPEI-1 towards SPEI-12.

Overall, the results showed a significant contribution of drought in changing river flow regimes across all the catchments of
the Indus Basin. The threshold (drought severity) increases with the time scale (SPEI-1 to SPEI-12) and from MIB to LIB.
Most of the catchments depicted severe drought as a threshold that causes ELF and LF at SPEI-6 and SPEI-12. The catchments
in LIB demonstrated extreme drought as a threshold at SPEI-12 that triggers ELF and LF events in the Indus Basin.

**4.3.2 Scenario 2: Time as a threshold variable**

Time is selected as a threshold variable to analyze the different time periods and their associated drought severity (SPEI) as
an independent variable. Each time period shows significant alterations where the river flows have almost similar
characteristics within a particular time period, i.e., no significant alterations in flow regimes within each time period. Drought
severity in tables 3, 4, and 5 represents the drought at a specific month, which separates one time period from another. Results
for SPEI-1 across the selected catchments of the Indus Basin are shown in Table 3. Threshold regression has divided most of
the catchments into four periods, where the drought severity differs from one-time period to another and catchment to
catchment. Most of the catchments in UIB depicted moderate drought as the drought severity, while the study duration (1980–
2018) is divided into three (Gilgit, Indus River at Shatial Bridge, and Tarbela catchments) and four (remaining basins of the
UIB) time periods. The coefficient in Table 3 quantifies the impact of drought on environmental flow. For instance, the
coefficient in Gilgit catchment shows that drought has a significant impact (0.949) on environmental flow during the period
of 1992-2011. During the mentioned period, Indus Basin has experienced frequent extreme drought events which not only
impacted the surface water availability but also other sectors including agriculture (Rahman et al., 2023). It should be noted
that the coefficient of SPEI-1 varies significantly from one catchment to another and from one period to another due to
significant variations in climatic and land use characteristics accompanied with frequent fluctuations in SPEI-1 estimates.

**Table 3.** Results of the threshold regression when time is used as threshold variable, where the study duration is divided into
different time periods and drought severity classes based on SPEI-1.

| Catchment | Time threshold | SPEI-1 | No. of period | Period | Coefficient | | Significance level | |
|---|---|---|---|---|---|---|---|---|
| | | | | | Constant | SPEI-1 | Constant | SPEI-1 |
| | 140 | Moderate | 1 | 1980-1991 | -0.071 | 0.507 | 0.003 | 0.000 |
| Gilgit | 384 | Moderate | 2 | 1992-2011 | 0.074 | 0.949 | 0.002 | 0.000 |
| | | | 3 | 2012-2018 | -0.14 | 0.661 | 0.092 | 0.000 |
| | 72 | Moderate | 1 | 1995-2000 | 0.275 | 0.802 | 0.000 | 0.000 |
| | 180 | Moderate | 2 | 2001-2009 | 0.189 | 0.592 | 0.000 | 0.000 |
| Hunza | 217 | Moderate | 3 | 2010-2012 | -0.174 | 1.103 | 0.019 | 0.000 |
| | | | 4 | 2013-2018 | 0.257 | 1.071 | 0.006 | 0.000 |
| | 150 | Moderate | 1 | 1980-1992 | -0.104 | 0.933 | 0.046 | 0.000 |
| Indus River | 216 | Moderate | 2 | 1993-1997 | 0.058 | 1.189 | 0.166 | 0.000 |
| at Bisham Qila | 359 | Moderate | 3 | 1998-2009 | 0.197 | 0.754 | 0.003 | 0.000 |
| | | | 4 | 2010-2018 | 0.147 | 0.747 | 0.007 | 0.000 |
| Indus River | 216 | Moderate | 1 | 1984-1997 | -0.076 | 0.778 | 0.003 | 0.000 |
| at Shatial Bridge | 347 | Moderate | 2 | 1998-2008 | 0.445 | 1.109 | 0.002 | 0.000 |
| | | | 3 | 2009-2014 | -0.060 | 0.456 | 0.015 | 0.000 |
| | 83 | Moderate | 1 | 1983-1989 | -0.311 | 0.847 | 0.002 | 0.000 |
| Indus River | 242 | Moderate | 2 | 1990-2002 | 0.284 | 0.748 | 0.006 | 0.000 |
| at Tarbela | | | 3 | 2003-2015 | -0.185 | 0.833 | 0.005 | 0.000 |
| | 146 | Moderate | 1 | 1980-1992 | -0.042 | 0.831 | 0.016 | 0.000 |
| Indus River | 271 | Severe | 2 | 1993-2002 | 0.275 | 0.861 | 0.004 | 0.000 |
| at Attock | 407 | Moderate | 3 | 2003-2013 | 0.112 | 1.241 | 0.005 | 0.000 |
| | | | 4 | 2014-2018 | -0.011 | 0.945 | 0.008 | 0.000 |
| | 277 | Moderate | 1 | 1980-2002 | -0.041 | 0.524 | 0.043 | 0.000 |
| Jhelum River | 408 | Moderate | 2 | 2003-2013 | 0.139 | 0.857 | 0.002 | 0.000 |
| | | | 3 | 2014-2018 | 0.076 | 0.915 | 0.006 | 0.000 |
| | 189 | Moderate | 1 | 1980-1995 | 0.089 | 0.974 | 0.008 | 0.000 |
| Kabul River | 290 | Severe | 2 | 1996-2003 | -0.016 | 0.909 | 0.010 | 0.000 |
| at Nowshehra | 407 | Moderate | 3 | 2004-2013 | -0.158 | 0.883 | 0.004 | 0.000 |
| | | | 4 | 2014-2018 | 0.021 | 0.735 | 0.012 | 0.000 |
| | 125 | Severe | 1 | 1980-1990 | 0.025 | 0.773 | 0.009 | 0.000 |
| Indus River | 201 | Severe | 2 | 1991-1996 | 0.021 | 0.545 | 0.007 | 0.000 |
| at Sehwan | 344 | Extreme | 3 | 1997-2008 | 0.012 | 0.917 | 0.015 | 0.000 |
| | | | 4 | 2009-2018 | 0.038 | 0.908 | 0.013 | 0.000 |

In contrast to other catchments in MIB, the Indus River at Attock and Kabul River at Nowshehra catchments depicted severe drought as a threshold for the period of 1993–2002 and 1996–2003, respectively. The river flow to the Indus River at Attock catchment depends on the outflow from Tarbela dam, where the outflow is extremely low during drought period. Similarly, river flow in the Kabul River is influenced by transboundary river issues between Afghanistan and Pakistan along with regional climate (arid climatic nature). Therefore, these catchments demonstrated severe drought as a threshold, where severe drought was observed during 1998–2002 in the history of Pakistan. The remaining catchments depicted moderate drought as a threshold in different time periods. On the other hand, the Indus River at Sehwan catchments depicted severe and extreme drought as a threshold in period 1/period 2 (1980-1990/1991-1996) and period 3 (1997-2008), respectively. Overall, the regression results of SPEI are significant at 1% levels in all the catchments.

Table 4 shows the results for SPEI-6, where study duration is divided into different time periods by considering time as a threshold variable. It should be noted that both the number of time periods and drought severity have increased significantly for SPEI-6 compared with SPEI-1. For instance, the number of time periods for the Gilgit catchment is five in the case of SPEI-6 as compared with three time periods in the case of SPEI-1. A similar increase in the number of time periods is observed for other catchments in UIB, MIB, and LIB. In addition to the increase in the number of time periods, the drought severity also increases where a severe drought corresponding to the time threshold is observed in almost all the catchments of UIB and MIB. Catchments in UIB depicted moderate drought across each individual time period as a threshold that separate one time period from another. The drought severity is highest in LIB among all the catchments of the Indus Basin, where the Indus River at Sehwan and Massan catchments depicted severe/extreme drought as a threshold. Jhelum River in MIB is divided into three distinct time periods where drought is of moderate severity. However, Indus River at Attock (dependent on the outflow from Tarbela) and Kabul River at Nowshehra (transboundary river catchment) catchments depicted both moderate and severe drought as a threshold to divide the study duration into different time periods. Overall, the results show more severe or extreme drought as an indicator in the pronounced drought periods, e.g., 1998–2002, 2007–2008, and 2012–2013. Table 4 shows that the SPEI coefficients are significant at 1% in all the catchments.

**Table 4.** Results of the threshold regression when time is used as threshold variable, where the study duration is divided into different time periods and drought severity based on SPEI-6.

| Catchment | Time threshold | SPEI-6 | No. of period | Period | Coefficient | | Significance level | |
|---|---|---|---|---|---|---|---|---|
| | | | | | Constant | SPEI-6 | Constant | SPEI-6 |
| | 159 | Moderate | 1 | 1980-1992 | 0.032 | 0.490 | 0.065 | 0.000 |
| | 276 | Severe | 2 | 1993-2002 | -0.160 | 0.904 | 0.009 | 0.000 |
| Gilgit | 337 | Moderate | 3 | 2003-2007 | 0.099 | 0.689 | 0.014 | 0.000 |
| | 400 | Moderate | 4 | 2008-2012 | 0.092 | 0.389 | 0.004 | 0.001 |
| | | | 5 | 2013-2018 | 0.105 | 0.767 | 0.004 | 0.000 |
| Hunza | 106 | Severe | 1 | 1995-2003 | -0.247 | 0.913 | 0.001 | 0.000 |

| | | | | | | | | |
|---|---|---|---|---|---|---|---|---|
| | 185 | Moderate | 2 | 2004-2009 | 0.161 | 0.631 | 0.001 | 0.000 |
| | 228 | Moderate | 3 | 2010-2013 | -0.106 | 0.312 | 0.004 | 0.000 |
| | | | 4 | 2013-2018 | -0.191 | 0.339 | 0.006 | 0.002 |
| Indus River at Bisham Qila | 72 | Moderate | 1 | 1980-1985 | 0.083 | 0.444 | 0.006 | 0.001 |
| | 215 | Moderate | 2 | 1986-1997 | -0.194 | 0.692 | 0.007 | 0.000 |
| | 273 | Severe | 3 | 1998-2002 | -0.157 | 0.915 | 0.004 | 0.000 |
| | 388 | Moderate | 4 | 2003-2012 | 0.191 | 0.716 | 0.001 | 0.000 |
| | | | 5 | 2013-2018 | 0.139 | 0.559 | 0.005 | 0.000 |
| Indus River at Shatial Bridge | 153 | Moderate | 1 | 1980-1996 | 0.089 | 0.504 | 0.012 | 0.000 |
| | 210 | Moderate | 2 | 1997-2001 | 0.044 | 0.756 | 0.009 | 0.000 |
| | 283 | Severe | 3 | 2002-2007 | -0.172 | 1.371 | 0.005 | 0.000 |
| | | | 4 | 2008-2014 | -0.165 | 0.884 | 0.003 | 0.000 |
| Indus River at Tarbela | 156 | Moderate | 1 | 1980-1995 | 0.146 | 0.508 | 0.002 | 0.000 |
| | 252 | Severe | 2 | 1996-2003 | 0.089 | 0.991 | 0.012 | 0.000 |
| | 364 | Severe | 3 | 2004-2013 | 0.037 | 1.214 | 0.032 | 0.000 |
| | | | 4 | 2013-2015 | -0.191 | 0.583 | 0.0010 | 0.000 |
| Indus River at Attock | 176 | Moderate | 1 | 1980-1994 | 0.038 | 0.652 | 0.014 | 0.000 |
| | 249 | Severe | 2 | 1995-2000 | 0.146 | 1.265 | 0.007 | 0.000 |
| | 387 | Severe | 3 | 2001-2012 | 0.158 | 0.926 | 0.004 | 0.000 |
| | | | 4 | 2013-2018 | 0.103 | 0.452 | 0.004 | 0.000 |
| Jhelum River | 204 | Moderate | 1 | 1980-1996 | 0.021 | 0.542 | 0.005 | 0.000 |
| | 339 | Moderate | 2 | 1997-2007 | 0.474 | 1.338 | 0.007 | 0.000 |
| | | | 3 | 2008-2018 | 0.057 | 0.381 | 0.013 | 0.000 |
| Kabul River at Nowshehra | 278 | Moderate | 1 | 1980-1994 | 0.075 | 0.612 | 0.009 | 0.000 |
| | 264 | Severe | 2 | 1995-2001 | -0.148 | 0.879 | 0.008 | 0.000 |
| | 346 | Severe | 3 | 2002-2008 | -0.105 | 0.996 | 0.008 | 0.000 |
| | 408 | Moderate | 4 | 2009-2013 | -0.155 | 0.603 | 0.009 | 0.000 |
| | | | 5 | 2014-2018 | 0.058 | 0.868 | 0.007 | 0.000 |
| Indus River at Sehwan | 123 | Severe | 1 | 1980-1990 | -0.131 | 0.728 | 0.007 | 0.000 |
| | 207 | Moderate | 2 | 1991-1997 | 0.108 | 0.646 | 0.031 | 0.000 |
| | 292 | Extreme | 3 | 1998-2004 | -0.347 | 1.592 | 0.001 | 0.000 |
| | 339 | Extreme | 4 | 2005-2008 | -0.225 | 0.938 | 0.000 | 0.000 |
| | | | 5 | 2009-2018 | -0.222 | 0.934 | 0.001 | 0.000 |

Table 5 represents the results for SPEI-12 where study duration is divided into different time periods by considering time as a threshold variable. The results show that SPEI-12 has the same number of time periods as SPEI-6 (across most of the catchments); however, the drought severity is increased significantly compared with SPEI-6. Moreover, the results are significant at the significance level of 1% for SPEI-12. Overall, the results show that catchments are vulnerable to severe and extreme drought events at SPEI-12 across the Indus Basin. For instance, the drought severity for catchments in UIB and MIB increases from moderate drought to severe drought; however, LIB depicted the severe drought as a threshold to divide the study period into different time periods.

**Table 5.** Results of the threshold regression when time is used as threshold variable, where the study duration is divided into different time periods and drought severity based on SPEI-12.

| Catchment | Time threshold | SPEI-12 | No. of period | Period | Coefficient | | Significance level | |
| --- | --- | --- | --- | --- | --- | --- | --- | --- |
| | | | | | Constant | SPEI-12 | Constant | SPEI-12 |
| | 136 | Moderate | 1 | 1980-1991 | -0.021 | 0.737 | 0.015 | 0.000 |
| | 212 | Severe | 2 | 1992-1997 | -0.192 | 0.873 | 0.005 | 0.000 |
| Gilgit | 338 | Severe | 3 | 1998-2007 | -0.148 | 0.784 | 0.009 | 0.000 |
| | 434 | Severe | 4 | 2008-2016 | -0.146 | 0.987 | 0.006 | 0.000 |
| | | | 5 | 2017-2018 | | | | |
| | 72 | Moderate | 1 | 1995-2000 | 0.275 | 0.802 | 0.000 | 0.000 |
| | 180 | Moderate | 2 | 2001-2009 | 0.189 | 0.592 | 0.000 | 0.000 |
| Hunza | 217 | Moderate | 3 | 2010-2012 | -0.174 | 1.103 | 0.019 | 0.000 |
| | | | 4 | 2013-2018 | 0.257 | 1.071 | 0.006 | 0.000 |
| | 62 | Moderate | 1 | 1980-1985 | 0.043 | 0.397 | 0.008 | 0.000 |
| | 207 | Severe | 2 | 1986-1997 | -0.119 | 0.955 | 0.007 | 0.000 |
| Indus River at Bisham Qila | 278 | Moderate | 3 | 1998-2003 | 0.024 | 0.401 | 0.016 | 0.005 |
| | 397 | Severe | 4 | 2004-2013 | -0.150 | 0.868 | 0.008 | 0.000 |
| | | | 5 | 2013-2018 | -0.055 | 0.791 | 0.013 | 0.000 |
| | 37 | Severe | 1 | 1984-1987 | -0.184 | 0.828 | 0.004 | 0.000 |
| Indus River at Shatial Bridge | 169 | Moderate | 2 | 1988-1998 | 0.087 | 0.722 | 0.013 | 0.000 |
| | 289 | Severe | 3 | 1999-2008 | -0.027 | 1.035 | 0.006 | 0.000 |
| | | | 4 | 2009-2014 | 0.075 | 0.735 | 0.007 | 0.000 |
| | 88 | Moderate | 1 | 1980-1991 | -0.051 | 0.582 | 0.008 | 0.000 |
| Indus River at Tarbela | 166 | Severe | 2 | 1992-1997 | -0.104 | 0.906 | 0.011 | 0.000 |
| | 292 | Severe | 3 | 1998-2008 | 0.072 | 1.138 | 0.010 | 0.000 |
| | 350 | Moderate | 4 | 2009-2012 | -0.056 | 0.845 | 0.016 | 0.000 |

| | | | | | | | | |
|---|---|---|---|---|---|---|---|---|
| | | | 5 | 2013-2018 | 0.044 | 0.606 | 0.013 | 0.000 |
| Indus River at Attock | 191 | Moderate | 1 | 1980-1996 | -0.051 | 0.485 | 0.009 | 0.000 |
| | 232 | Severe | 2 | 1997-2000 | -0.240 | 1.422 | 0.005 | 0.000 |
| | 337 | Severe | 3 | 2001-2008 | -0.141 | 0.752 | 0.002 | 0.000 |
| | 376 | Severe | 4 | 2009-2012 | 0.042 | 0.985 | 0.013 | 0.000 |
| | | | 5 | 2013-2018 | 0.018 | 0.748 | 0.007 | 0.000 |
| Jhelum River | 195 | Moderate | 1 | 1980-1996 | 0.064 | 0.647 | 0.007 | 0.000 |
| | 263 | Severe | 2 | 1997-2002 | -0.143 | 1.185 | 0.003 | 0.000 |
| | 395 | Moderate | 3 | 2003-2013 | -0.046 | 0.894 | 0.012 | 0.000 |
| | | | 4 | 2014-2018 | -0.038 | 1.123 | 0.014 | 0.000 |
| Kabul River at Nowshehra | 87 | Moderate | 1 | 1980-1987 | 0.016 | 0.664 | 0.015 | 0.000 |
| | 265 | Severe | 2 | 1988-2002 | -0.199 | 0.819 | 0.002 | 0.000 |
| | 397 | Severe | 3 | 2003-2013 | 0.036 | 0.929 | 0.013 | 0.000 |
| | | | 4 | 2017-2018 | -0.083 | 0.942 | 0.009 | 0.000 |
| Indus River at Sehwan | 188 | Severe | 1 | 1980-1987 | 0.066 | 0.912 | 0.015 | 0.000 |
| | 283 | Extreme | 2 | 1988-1995 | -0.147 | 1.458 | 0.008 | 0.000 |
| | 428 | Severe | 3 | 1996-2007 | -0.029 | 0.904 | 0.014 | 0.000 |
| | 499 | Extreme | 4 | 2008-2013 | -0.281 | 1.528 | 0.004 | 0.000 |
| | | | 5 | 2014-2018 | -0.134 | 0.758 | 0.007 | 0.000 |

Generally, the results show that environmental flow can be divided into different time periods, where drought severity varies from one time period to another and from SPEI-1 to SPEI-12. For instance, SPEI-1 showed moderate drought as a threshold that divided the study duration into different periods across different catchments. The drought severity increases to severe drought in most of the catchments when SPEI-12 is considered as an independent variable. Moreover, the catchments in MIB depicted relatively lower vulnerability to drought compared with those in UIB and LIB. Besides the climate-induced impacts on river flow, anthropogenic activities and transboundary river issues further worsen the impact of climate on ELF and LFs.

## 4.4 Hydrological alterations in the Indus Basin

RVA is mostly used to analyze the hydrological alterations in flow regimes by comparing the flow in pre-impact period against the post impact period. In this study, we used whole period (1980-2018) as a pre-impact period and the specific drought years as a post-impact period to assess the impact of drought on environmental flow. HAF is calculated from the results of RVA and is spatially distributed to demonstrate the hydrological alterations in the Indus Basin for 18 EFC components. The selected EFC components are related to low environmental flow (i.e., ELF and LF) during the drought period, which is calculated at the catchment scale. Fig. 7 demonstrates that most of the catchments in the Indus Basin are subjected to high alterations during

most months of the year except August and September dominated by moderate alternations. Overall, environmental flow in the catchments of UIB is comparatively less vulnerable to drought compared with catchments in LIB. Further, low vulnerability (moderate alterations) is observed in most of the catchments of Indus Basin during the monsoon season (July–September), during which Pakistan receives the most intense precipitation with a magnitude of 55%-60% of the annual precipitation (Dimri

et al., 2015). The monsoon precipitation contributes to irrigate most of the irrigation areas with approximately 30 billion m$^3$ of water (Rahman et al., 2022b). High precipitation results in no or moderate drought events during the monsoon season. Besides, flow is also relatively high in the monsoon season due to relatively high temperature that accelerates the snow and glacier melting process in UIB (Hasson et al., 2017). Therefore, hydrological alterations in the Indus Basin are comparatively lower in monsoon season compared with other seasons. The alterations increase from monsoon to post-monsoon (October–November)

and winter (December–March) seasons. During the winter season, except for March, most of the catchments depicted high alterations due to moderate precipitation and relatively low flow in the rivers (Archer, 2003; Sharif et al., 2013).

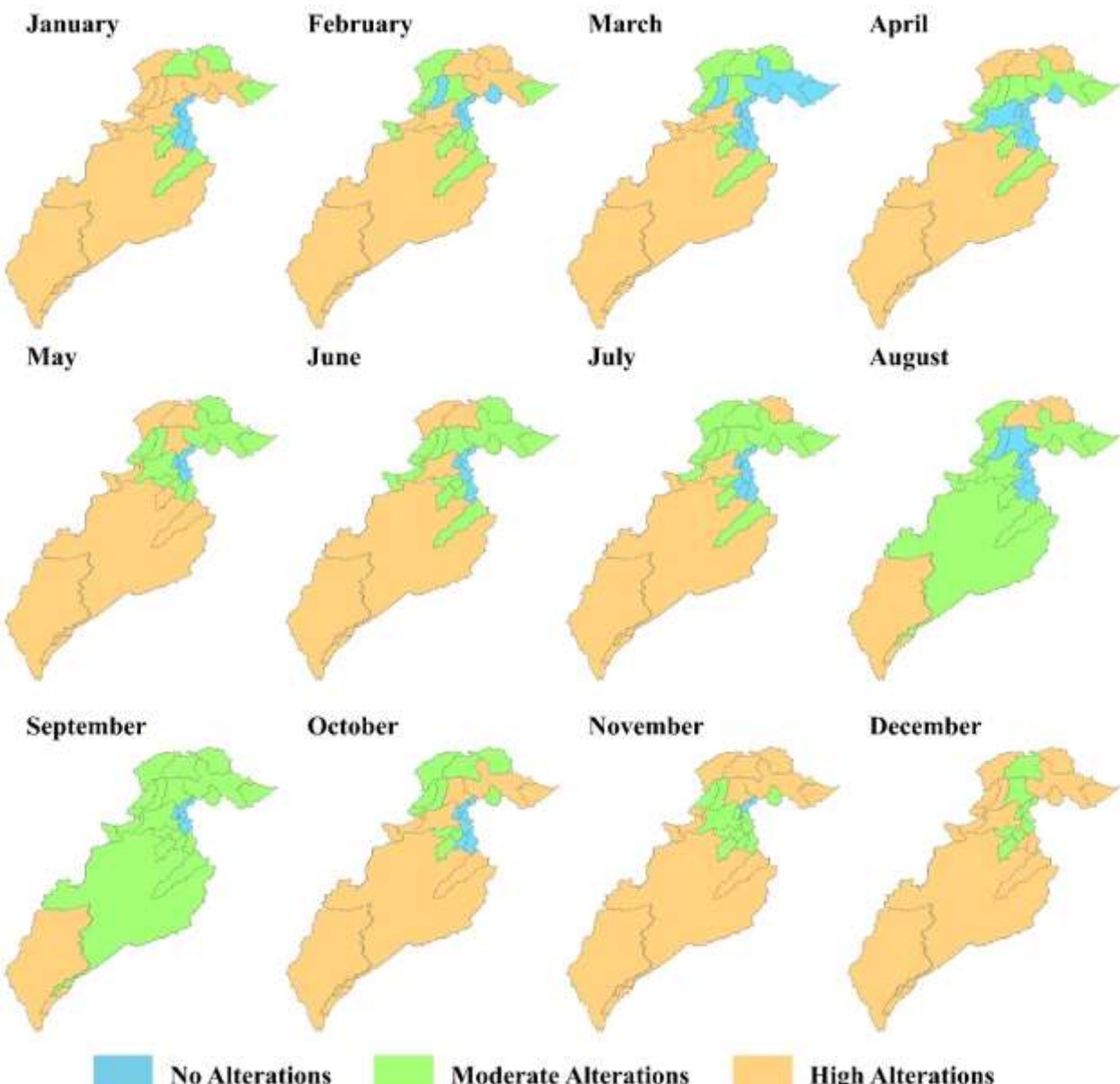

**Figure 7.** Hydrological alterations in environmental flow due to drought at a monthly scale averaged over the entire study period in the Indus Basin

On the basis of geographical division of the Indus Basin, most of the catchments in UIB depicted high to moderate alterations in different months (Fig. 7). Hunza, Gilgit and Chitral catchments are experiencing high alterations in most of the months as compared to the remaining catchments of UIB. The river flows in glacial regions are extremely seasonal, i.e., minimum flow in the winter period due to snow accumulation and a relatively pronounced melting in the summer period (Huss and Hock,

2018), especially in the UIB (Khan et al., 2020). However, the contribution of glacier melt to river flow is decreasing due to intense precipitation (Bashir et al., 2017). "Karakoram Anomaly" is defined as glaciers in the western Karakoram, eastern Hindukush, and northwestern Himalayan Mountain ranges are not responsive to global warming in the same pattern as their counter parts (Bashir et al., 2017). In other words, the rates of their retreat are usually less than the global average, where some of the glaciers are stable or increasing. Therefore, this local phenomenon may further contribute to high alterations in ELF and LF events in UIB. Climate change is one of the prominent factors that can further intensify both low flow or high flow events. River flows in the Indus Basin depends on the snowmelt from UIB; thus, most of the catchments in MIB and particularly LIB depicted high alterations during different seasons. The eastern catchments of MIB receive comparatively more precipitation than the western catchment and thus depicted no significant alterations in most of the months, except the winter season (November and December).

LIB is most vulnerable to drought due to low precipitation and high temperature, and thus high hydrological alterations due to drought are observed at Indus River at Massan and Sehwan catchments (Fig. 7). Besides the local changes in climate, water withdrawal from the Indus River system to IBIS for irrigation purpose has a significant contribution to the high vulnerability of LIB catchments. The Indus River system, comprised of eastern and western rivers along with their tributaries, has an annual average runoff of approximately 180 billion cubic meters (BCM), out of which 128 BCM is diverted to the IBIS to irrigate approximately 22.14 million ha area (Basharat, 2019). Therefore, the impact of drought on environmental flow is further intensified in LIB due to such a huge amount of water diversion. Overall, the seasonal evaluation showed that catchments in the Indus Basin have moderate alterations in the monsoon season. Further, catchments in MIB and parts of UIB are less vulnerable to drought as compared to LIB.

Other EFC components considered in this study include ELF and LF events at 1-day, 7-day, 30-day, 90-day, low pulse count, and low pulse duration (Fig. 8). The results show increased alterations with the increase in cumulative time. For instance, most of the catchments depicted no alterations (UIB and MIB) at 1-day minimum EFC, which increases gradually to high alterations with the increase in accumulated time (30-day and 90-day minimum). On the other hand, alterations in low pulse count are moderate in most of the catchments of UIB, no alterations in eastern catchments of MIB, and high alterations in the remaining catchments of MIB and LIB. On the contrary, the results show that Hunza, Gilgit, and Chitral catchments in UIB have high alterations in terms of low pulse duration. In other words, these catchments have persistent ELF and LF events for an extended period of time. The remaining catchments of UIB and most of the catchments in MIB (except the arid/hyper-arid regions) depicted moderate alterations in terms of the duration of low pulses. Similar to low pulse count, catchments in LIB depicted high alterations in low pulse duration.

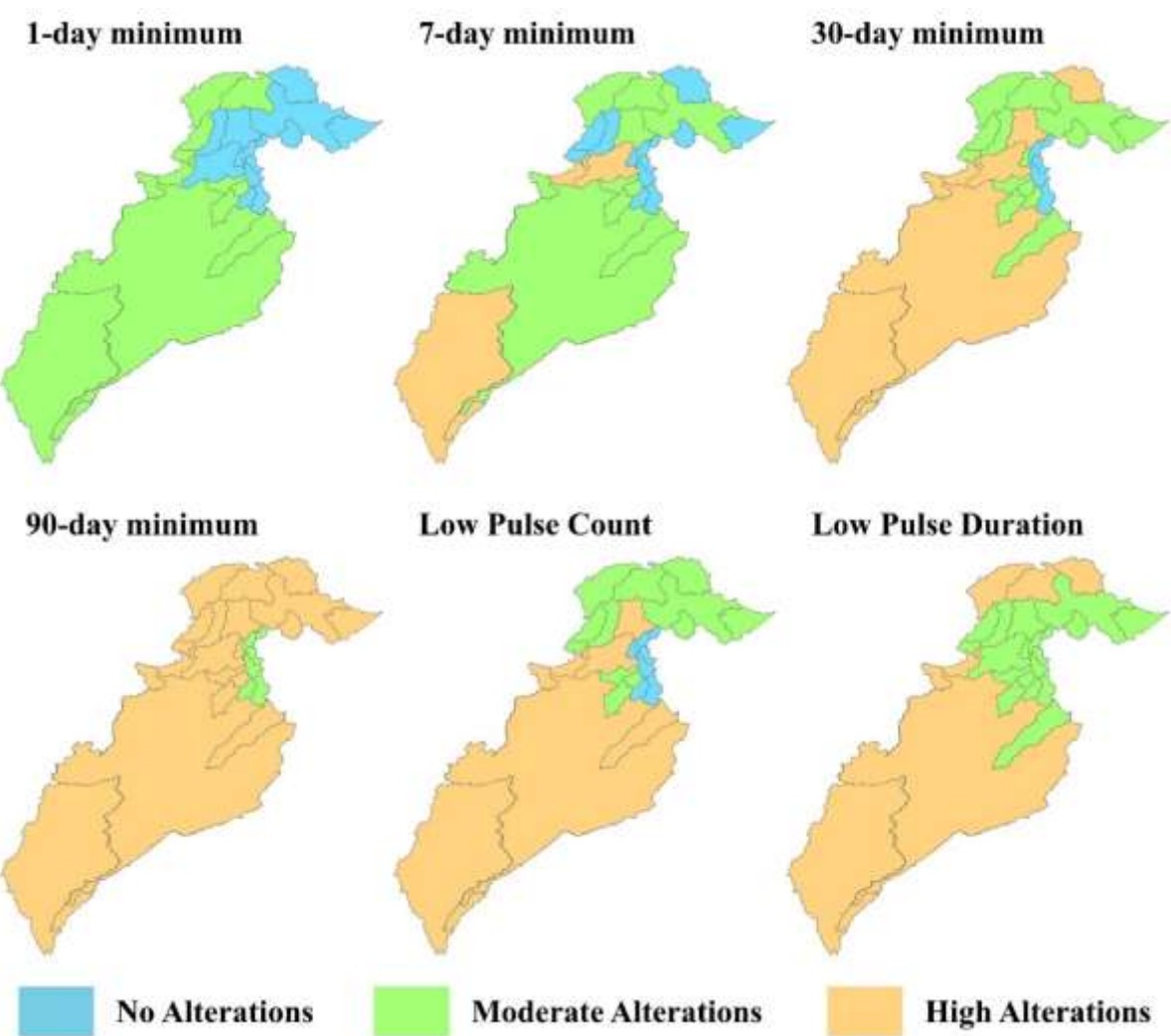

**Figure 8.** Hydrological alterations in the Indus Basin

**5 Discussion**

Pakistan has been added to the list of water-stressed countries due to water scarcity issues under severe climate change and land use change scenarios. However, it is relatively difficult to precisely assess the impact of climate change on water availability in the Indus Basin because of uncertainties due to topographic complexity, local changes in climate that influence the natural glacial and snow melt process, glacial retreat and shifts in precipitation pattern (Janjua et al., 2021). The UIB 490 contributes approximately 45% of the flow to the main rivers in Indus Basin, suggesting the high vulnerability of glacial melt to climate change and results in a 40% of surge in riverine flow (Janjua et al., 2021). However, on the long run, the average flows in the main tributaries of the Indus Basin are reduced by almost 60% (Briscoe and Qamar, 2006). This reduction in river

flow is mainly associated with the global warming, i.e., the evapotranspiration is likely to increase significantly in the irrigated areas of the Indus Basin resulting in the increase of water demand for irrigation (National Research Council, 2012). The Indus Basin (Pakistan) receives highest magnitude of precipitation (50%-60%) during the monsoon season that results in approximately 85% of the annual discharge in the Indus Basin, which will be significantly altered in a couple of decades due to climate change (National Research Council, 2012).

Extreme events, i.e., droughts and floods resulted due to climate change has tested the inhabitants of the Indus Basin in a number of ways. Pakistan is an agricultural country, where the economic development of Pakistan depends on sustainable agricultural production (Rahman et al., 2023b). Besides the direct impact of droughts on agricultural productivity, the droughts also cause significant reduction in surface water availability and consequently the irrigation water supply. The estimated water consumption by municipal and industrial sectors in Pakistan is approximately 5.3 km$^3$, which is projected to increase to 14 km$^3$ by 2025 (Condon et al., 2014). Therefore, there will be limited available water for irrigation purpose and will significantly impact the water availability in rivers and in turn the sustainable EFs.

The Indus Basin Irrigation System (IBIS) irrigates approximately 150,000 km$^2$ out of 190,000 km$^2$ of cultivated crop area in the Indus Basin (Ahmad, 2005), resulting in the deterioration of environmental water and the Indus delta ecosystem because of lack of sustainable minimum flow in the riverine system (Janjua et al., 2021). The conditions required for minimum flow in rivers becomes more critical during the drought periods; for instance, the difference between water demand and supply was 20% during the 2000–2002 drought period (Briscoe, 2006). Keeping in view the worse condition of EF in the Indus Basin, it was suggested by the experts in 2005 that we should sustain a minimum of 141.58 m3/sec flow in river at Kotri Barrage to the sea (González et al., 2005). Due to the extensive withdrawal of surface water from the rivers by the IBIS, it was decided to ensure a 30 km$^3$ of cumulative flow for a period of 5 years in the Indus River (González et al., 2005).

Beside the water withdrawal through IBIS, drought has significant contributions in reducing the flow in rivers of the Indus Basin (Rahman et al., 2023a). The persistent meteorological drought reduces the water availability in river flows, which then ultimately translates into insufficient release of EF (Pena-Guerrero et al., 2020). The frequency and intensity of drought in the Indus Basin has been increased substantially in the recent decades, which resulted in high variability in meteorological and hydrological droughts. Rahman et al. (2023a) propagated drought from one catchment to another in a systematic approach using the principal component analysis (PCA) to understand the variability in both meteorological and hydrological droughts. Results showed high variability in hydrological droughts compared to meteorological droughts in most of the catchments in Indus Basin. In other words, most of the catchments experience a decrease in river flow associated with meteorological drought and thus depicting that drought is one of the major threats to sustainable ecosystem and EF.

This study is first of its kind that evaluated the impact of drought on EF under two distinct scenarios using threshold regression: i) drought severity that causes LFs and ELFs in the rivers, and ii) the months where drought caused LF and ELF. Keeping in view the importance of maintaining minimum flow in rivers and frequent severe drought events in the Indus Basin, the relationship between drought and EF in the Indus Basin should further be investigated in more details. More investigations would be also required on how other extreme events, flood for instance, affect the environmental flow. This study is conducted

across a data scarce region; therefore, we used PCA to estimate and propagate drought from one catchment to another. Although, the PCA algorithm retains maximum variance and keeps the original structure of the data of a catchment under investigation, there might exists some uncertainties in the propagated data. Therefore, it is recommended to use dense in-situ data or use continuous data (retrieved from remote sensing products) for the sake of comparison and detailed investigation. Besides, the threshold regression failed to identify thresholds for specific events (i.e., extreme low flow and low flow). It would be interesting to use machine learning techniques to determine drought severity that causes extreme low flows and low flows in the future.

**6 Conclusion**

In this study, the impact of drought on environmental flow in 27 catchments of the Indus Basin is assessed using the indicators of hydrologic alteration (IHA). The standardized precipitation evapotranspiration index (SPEI) is used to calculate drought from the systematically propagated principal components of precipitation and temperature estimated using principal component analysis (PCA). Threshold regression is used to identify a specific drought severity and month that trigger the low flows. In addition, range of variability analysis (RVA) is used to quantify the impact of drought on extreme low flows. The RVA results are also used to calculate the Hydrological Alteration Factor (HAF), which indicates the category of alterations (no alteration, moderate and high alterations) in each catchment. The main conclusions are:

(1)  Most of the catchments in Indus basin showed persistent drought events during the periods 1984 to 1986, 1991/1992, 1997 to 2003, 2007 to 2008, 2012 to 2013, and 2017 to 2018. The drought is evident on a larger time scale, i.e., SPEI-12 compared to SPEI-6 and SPEI-1. Moreover, the drought is more severe in the Lower Indus Basin (LIB) than in the Upper Indus Basin (UIB). The analyses have shown that temperature plays a crucial role in the occurrence of droughts. In addition, local climate, topography, length of period, and seasonality contribute significantly to drought variability.

(2)  The distribution of Environmental Flow Components (EFCs) shows a significant decrease in river flow during drought years. The magnitude of extreme low flow (ELF) and low flow (LF) is low in the UIB, while it increases significantly toward LIB. In the transboundary river catchments, significant changes are observed in the ELF and LF events. Overall, the magnitude and frequency of the ELF and LF events increase with the increase in drought severity.

(3)  Threshold regression results (Scenario 1, where drought severity is considered the threshold variable) showed that most of the catchments were affected by moderate drought at shorter time scales (SPEI-1 and SPEI-6). However, at longer time scales (SPEI-12), the threshold of drought severity increases to severe and extreme drought. The drought severity threshold is highest at LIB at all time scales. Catchments in the MIB (eastern catchments) are mainly influenced by human-induced activities, while the changes in river flow across the UIB and western MIB are triggered by climate-induced activities such as drought. Similar observations apply to LIB, where catchments are mainly influenced by climatic factors.

(4)  Scenario- 2 (where time is considered as a threshold variable) provided a clear insight into the impact of drought on environmental flow by dividing the study duration into different time periods characterized by different characteristics, i.e.,

significant alterations in flow regime between the different time periods and almost similar characteristics in each one considering the severity of the drought. The study duration is divided into three to five time periods where moderate to severe drought triggered ELF and LF in most of the catchments. Drought severity increases from moderate in the (UIB/MIB) to extreme in the (LIB), and this increase is associated with the increase in time scale from SPEI-1 to SPEI-12.

(5)   Threshold regression analysis was useful in quantifying alterations in environmental flow due to drought. LIB experienced significant alterations in environmental flow as compared to UIB and MIB. In addition, the SPEI coefficient from threshold regression in scenario-2 (shown in Tables 3 and 4) increases with increase in drought severity, suggesting that SPEI has a significant impact on environmental flow in specific catchments.

(6)   Most of the catchments were subject to high alterations in all months of the year. Drought impacts on environmental flow are more severe in LIB, followed by UIB, than in MIB. Climate change, topography, land use, and anthropogenic activities have significant impacts on the environmental flow. For example, moderate or no alterations are observed during the monsoon season, while high alterations occur in winter. In addition to seasonal variations in river flow, temperature plays a critical role in variability of drought and its impact on environmental flow. The Karakoram anomaly is one of the key factors contributing to high alterations in ELF and LF events in the UIB and thus in MIB and LIB.

Understanding the impact of climate-induced changes (especially droughts) on environmental flow is extremely important to ensure the minimum flow required to maintain ecosystem services. This study provided detailed insights into changes in environmental flow with changes in drought severity that will serve as a useful guide for researchers, government organizations, policy makers, and local authorities to reconsider decisions in light of climate change impacts on environmental flow.

*Code availability*: This study used the freely available code/package, i.e., SPEI in R to calculate the drought and Principal Component Analysis (PCA) to propagate drought from one catchment to another. Moreover, IHA software is used to calculate the EFCs and perform RVA analysis.

*Data availability*: Data is available on request from the first author (khalil628@tsinghua.edu.cn).

*Author contribution*:  KUR conceptualized the idea, developed the methodology, conducted the analysis, validated the results, and wrote the original draft paper; SS and KSB contributed to conceptualization, analysis, methodology, and validation and supervised the research; HFG, KZJ, and KZ were involved in the data curation and methodology. All authors were involved in reviewing and editing the paper.

*Competing interests*: The authors declare no competing interests.

*Acknowledgments*: This research was supported by the National Natural Science Foundation of China (Grant number 51839006 and 52250410336), China Postdoctoral Science Foundation (Grant number 2022M721872), and the Shuimu Scholar Program of Tsinghua University (Grant number 2020SM072). This research was partially supported by the Higher Education

Commission (HEC), Pakistan under the project CPEC-161. We are grateful to Pakistan Meteorology Department (PMD) and
Water and Power Development Authority (WAPDA) for providing the meteorological and hydrological data.

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
