# Peer review of "Catchment scale assessment of drought impact on environmental flow in the Indus Basin, Pakistan"

_Natural Hazards and Earth System Sciences, 2023_

## Author Comment (AC1)

This is a comprehensive manuscript which discuss the environmental flows and droughts in the entire Indus basin in Pakistan. Overall manuscript is well written, figures and results are well presented and conclusions are valuable. However, the quality of manuscript needs to be further improved, here are few comments which may be useful in this regards.

Authors are thankful to the reviewer for the positive feedback and valuable comments. All the comments are considered carefully, and the manuscript are revised accordingly. Response to each comment is given below in blue color.

1. In abstract section authors concluded " The alterations are subject to several factors, including climate change, seasonality of the river flow, land use changes, topography, and anthropogenic activities" which type of analysis have been performed to reach these conclusions which seems to be generic.

**Response:** This conclusion has been made based on the results from our recently paper under review. However, no such analyses are carried out in this manuscript to support this claim. Therefore, the statement has been removed from the abstract.

2. In the Introduction section authors stated "The Indus River basin is one of the typical basins facing substantial climate and land use changes, resulting in limited water availability". However, no references have been added to support this statement authors should state which part of the Indus basin has faced serious land use and climate changes?

**Response:** References are added to support this statement.

"The Indus River basin is one of the typical and most depleted basins due to substantial climate and land use changes, resulting in limited water availability (Azmat et al., 2019; Immerzeel et al., 2010; Laghari et al., 2012; Sharma et al., 2010). Upper Indus Basin (UIB) is the hotspot for climate change, whereas Middle Indus Basin (MIB) and Lower Indus Basin (LIB) are dependent on the availability of water from UIB. Several studies have reported an increase in future precipitation and temperature (Forsythe et al., 2014; Nepal and Shrestha 2015; Rajbhandari et al., 2015); however, Shahid and Rahman (2021) reported that the findings in most of the studies are not consistent with global trends due to a number of reasons".

*Azmat, M., Wahab, A., Huggel, C., Qamar, M.U., Hussain, E., Ahmad, S. and Waheed, A.: Climatic and hydrological projections to changing climate under CORDEX-South Asia experiments over the Karakoram-Hindukush-Himalayan water towers. Sci. Total Environ., 703, p.135010, 2019.*

*Forsythe, N.; Fowler, H.J.; Blenkinsop, S.; Burton, A.; Kilsby, C.G.; Archer, D.R.; Harpham, C.; Hashmi, M.Z.: Application of a stochastic weather generator to assess climate change impacts in a semi-arid climate: The Upper Indus Basin. J. Hydrol. 517, 1019–1034, 2014.*

*Immerzeel, W.W., Van Beek, L.P. and Bierkens, M.F.: Climate change will affect the Asian water towers. Sci., 328(5984), pp.1382-1385, 2010.*

*Laghari, A.N., Vanham, D. and Rauch, W.: The Indus basin in the framework of current and future water resources management. Hydrol. Earth Sys. Sci., 16(4), pp.1063-1083, 2012.*

*Nepal, S., Shrestha, A.B.: Impact of climate change on the hydrological regime of the Indus, Ganges and Brahmaputra river basins: a review of the literature. Int. J. Water Res. Devel. 31, 201–218, 2015.*

*Rajbhandari, R.; Shrestha, A.B.; Kulkarni, A.; Patwardhan, S.K.; Bajracharya, S.R.: Projected changes in climate over the Indus river basin using a high resolution regional climate model (PRECIS). Clim. Dyn. 44, 339–357, 2015.*

*Shahid, M. and Rahman, K.U.: Identifying the annual and seasonal trends of hydrological and climatic variables in the Indus Basin Pakistan. Asia-Pacific J. Atmos. Sci., 57, pp.191-205, 2021.*

*Sharma, B., Amarasinghe, U., Xueliang, C., de Condappa, D., Shah, T., Mukherji, A., Bharati, L., Ambili, G., Qureshi, A., Pant, D. and Xenarios, S.: The Indus and the Ganges: river basins under extreme pressure. Water Int., 35(5), pp.493-521, 2010.*

3. How did the authors categorize the flow data into extreme low flow and low flow?

**Response:** The Indicators of Hydrologic Alterations (IHA) usually consists of a total 67 parameters, which are grouped into IHA parameters (33) and Environmental Flow Components (EFCs) (34). The EFCs are grouped into five major categories, (i) low flows, (ii) extreme low flows, (iii) high flow pulses, (iv) small floods, and (v) high floods. The detailed information about the IHA parameters and EFCs are available in user manual (*https://www.conservationgateway.org/Documents/IHAV7.pdf*).

Since the flow in rivers are minimum during the drought period; therefore, this study considered on the extreme low flow (ELF) and low flow (LF) components from the five major EFC classes to understand the impact of drought (i.e., threshold that triggers the ELF and LF) on EF.

4. Various data qualities issues have been reported regarding the hydrological and meteorological datasets of the Indus basin. How authors addressed the missing datasets and which type of analysis have been performed to check the data quality?

**Response:** Data in Pakistan (Indus Basin) is usually manually collected by Pakistan Meteorology Department (PMD) and Water and Power Development Authority (WAPDA). Therefore, the collected data has several issues, including errors due to personal and instrumental errors, splashing due to climate, errors due to winds, topography, etc. These errors result in poor quality and missing data. The initial attempts are made by PMD and WAPDA to rectify the data following the standard code of WMO-N issued by the World Meteorological Organization. Besides, we have also performed data quality tests including the kurtosis and skewness methods to check the data quality, and the missing data is filled by zero-order methods following Rahman et al. (2018).

5. On which basis authors have done the demarcation of the indus basin into UIB, MIB and LIB?

**Response:** The demarcation of the Indus Basin into UIB, MIB, and LIB is done following Aftab et al., (2022), Rajbhandari et al., (2015), and Shahid et al., (2021).

*Aftab, F., Zafar, M., Hajana, M.I. and Ahmad, W.: A novel gas sands characterization and improved depositional modeling of the Cretaceous Sembar Formation, Lower Indus Basin, Pakistan. Front. Earth Sci., 10, p.1039605, 2022.*

*Rajbhandari, R.; Shrestha, A.B.; Kulkarni, A.; Patwardhan, S.K.; Bajracharya, S.R.: Projected changes in climate over the Indus river basin using a high resolution regional climate model (PRECIS). Clim. Dyn. 44, 339–357, 2015.*

*Shahid, M., Rahman, K.U., Haider, S., Gabriel, H.F., Khan, A.J., Pham, Q.B., Mohammadi, B., Linh, N.T.T. and Anh, D.T.: Assessing the potential and hydrological usefulness of the CHIRPS precipitation dataset over a complex topography in Pakistan. Hydrol. Sci. J., 66(11), pp.1664-1684, 2021.*

6. Various studies have been performed to understand the drought in the Indus basin Authors can open the scholar and search from key word droughts in Indus basin. However, no discussion has been performed to compare the results of this study with literature. Discussing the results with previous studies will be useful for readers and this manuscript has potential to be extended for a brief discussion.

**Response:** A detailed discussion section is added to the manuscript following the recommendations of reviewer. However, there is no such studies available in literature that identified the drought severity causing low flow and extreme low flow in the rivers, and quantified the impact of drought on environmental flow:

Pakistan has been added to the list of water-stressed countries due to water scarcity issues under severe climate change and land use change scenarios. However, it is relatively difficult to precisely assess the impact of climate change on water availability in the Indus Basin because of uncertainties due to topographic complexity, local changes in climate that influence the natural glacial and snow melt process, glacial retreat, and shifts in precipitation pattern (Janjua et al., 2021). The UIB contributes approximately 45% of the flow to the main rivers in Indus Basin, suggesting the high vulnerability of glacial melt to climate change and results in a 40% of surge in riverine flow (Janjua et al., 2021). However, on the long run, the average flows in the main tributaries of the Indus Basin are reduced by almost 60% (Briscoe and Qamar, 2006). This reduction in river flow is mainly associated with the global warming, i.e., the evapotranspiration is likely to increase significantly in the irrigated areas of the Indus Basin resulting in the increase of water demand for irrigation (National Research Council, 2012). The Indus Basin (Pakistan) receives highest magnitude of precipitation (50%-60%) during the monsoon season that results in approximately 85% of the annual discharge in the Indus Basin, which will be significantly altered in a couple of decades due to climate change (National Research Council, 2012).

[revised manuscript text omitted]

7. I have some minor comments regarding use of abbreviations which are unnecessary, should be reduced and must be explained at first use e.g. In abstract section authors should explain EFs before first use.

**Response:** Corrected as suggested. The term EF is already explained in the first line of the abstract before its use.

---

## Author Comment (AC2)

**General comments:**

In this study, Rahman et al. explore the impact of drought on environmental flow (EF) in 27 catchments in the Indus basin, focusing on the period 1980-2018, using the Indicators of Hydrologic Alterations (IHA). The authors use SPEI to quantify drought at various timescales. Drought impact on low EFs is quantified using RVA. Their results show that the Lower Indus Basin (LIB) is more vulnerable to drought than the Upper Indus Basin (UIB) and that drought is related to extreme low flow (ELF) and low flow (LF).

The study on drought is in scope of the NHESS journal and is a good contribution to the field of drought and environmental flow. The figures are well presented. However, the method and analysis need some elaboration, especially the explanation on the IHA. The text is well written although some paragraphs can be shortened because of repetition. The Discussion need some elaboration to discuss the findings with existing literature. Below find my comments.

The authors are thankful to the reviewer for the careful revision and insightful comments, which has significantly improved the manuscript readability and the quality. The manuscript is revised following the comments from reviewer and response to each comment is given (in blue) in the response file.

**Specific comments:**

  • **Abstract**

The abstract is very concise, I suggest to provide a bit more information on the Indicators of Hydrologic Alterations (IHA) in relation to the Range of variability analysis (RVA). In addition, line 29-30 sound a bit too generic and is not directly studied here, please be more specific.

**Response:** Thank you for the valuable comments. The abstract is revised as suggested.

  a. Threshold regression is used to determine the severity of drought (scenario-1, drought severity that causes low flows)) and month (scenario-2, months where drought has resulted in low flows) that trigger low flows in the Indus Basin.
  b. The impact of drought on low EFs is quantified using Range of variability analysis (RVA). RVA is an integral component of IHA, which is mainly used to study the hydrological alterations in environmental flow components (EFCs) by comparing the pre- and post-impact periods of human and climate interventions in EFCs.

The original Lines 29-30 have been removed.

  1. **Introduction**

Line 33) "to the quality, timing and quality of freshwater flows": The word "quality" is repeated.

**Response:** Corrected. Thank You for highlighting the mistake.

Line 37-39) The authors mention here that 65% of the discharge in rivers poses a moderate to severe threat to biodiversity; in what way, in relation to water quantity or quality? In addition, since when are those numbers altered? Please be more specific.

**Response:** The statement is revised as "On a global scale, it is estimated that approximately 65% of the discharge and the supported aquatic habitat is under moderate to high threat (Vörösmarty et al., 2010), connectivity of 48% of rivers is diminished (Grill et al., 2019), and fish biodiversity has been significantly altered in 53% of the rivers (Su et al., 2021).

The research was published in 2010, where the authors (Vörösmarty et al., 2010) studied threats to water security and biodiversity due to anthropogenic activities. There is no specific time frame mentioned in the study to understand that temporal variations in magnitude of global river discharge.

*Vörösmarty, C.J., McIntyre, P.B., Gessner, M.O., Dudgeon, D., Prusevich, A., Green, P., Glidden, S., Bunn, S.E., Sullivan, C.A., Liermann, C.R. and Davies, P.M.: Global threats to human water security and river biodiversity. Nature, 467(7315), pp.555-561, 2010.*

Line 46) "**the** alterations in flow regime": which alterations?

**Response:** The statement is revised as "the hydrological alterations in flow regime".

Line 55-56) "Eckstein et al. (2018) that …": this part does not fit into the sentence, what is meant here?

**Response:** The statement is revised as "Pakistan (Indus Basin) is highly vulnerable to climate change and placed at 8th position among the countries most affected by climate change (Eckstein et al., 2018). Therefore, the Indus Basin experienced more frequent and severe extreme events in the recent few decades."

Line 58) Note that soil moisture drought is more specific than agricultural drought, see Van Loon (2015).

**Response:** It is quite right that soil moisture drought is more specific than agricultural drought, and agricultural drought is a phenomenon that occurs when there is insufficient moisture in the soil to support crop growth. In many references, drought is broadly classified into meteorological, hydrological, agricultural, and socio-economic droughts. As agricultural drought / soil moisture deficit is not studied in this manuscript, this sentence was kept in the manuscript without revision.

Line 70) Where refers "this" to?

**Response:** Thank You for pointing out the mistake. The statement is corrected as "this study for the first time evaluated the impact of drought on EF using the Indicators of Hydrologic Alterations (IHA)".

1. **Study area**

Figure 1) Where do the colors refer to in Figure 1d?

**Response:** The figure is revised into a single color consistent with the remaining colors.

Line 96-97) This is mentioned before already in the same section.

**Response:** The lines are removed as suggested.

In general, there is some overlap in text between the Introduction and Study area as the Indus basin has been addressed in the Introduction already. I suggest to better align those texts to avoid overlap and to address the urgency and the knowledge gap of studying EF in relation to drought in the Introduction.

**Response:** the repeating statements have been removed from the introduction and study area sections.

1. **Methodology**

Line 141) Please provide more information about using PCA in this regard? Has it done before in this way etc.?

**Response:** Yes, the authors (Rahman et al., 2023a) have used the PCA method to systematically propagate drought from one catchment in the Indus Basin to another catchment. To avoid the repetition, we have added the reference in line 143 in the revised version of the manuscript.

*Ur Rahman, K., Shang, S., Balkhair, K. and Nusrat, A.: Catchment-Scale Drought Propagation Assessment in the Indus Basin of Pakistan Using a Combined Approach of Principal Components and Wavelet Analyses. J. Hydrometeorol., 24(4), pp.601-624, 2023a.*

Line 180-181) How are the IHA used to compute the EFC and how are ELF and LF defined?

**Response:** Thank you for your valuable comment. The application of IHA mainly depends on its calibration procedure. In this study, the IHA is calibrated using advanced calibration technique following the guidelines in the use manual (*https://www.conservationgateway.org/Documents/IHAV7.pdf*) to capture both high flows and low flows, which is already mentioned in the manuscript (Lines 182-188). After calibrating the IHA's EFC parameters with the observed input data, we selected only the ELF and LF because ELF and LF are only observed during the drought period.

Line 184) What are the EFC parameters?

**Response:** The Indicators of Hydrologic Alterations (IHA) usually consists of a total 67 parameters, which are grouped into IHA parameters (33) and EFCs (34). The EFC parameters are grouped into five major categories, (i) low flows, (ii) extreme low flows, (iii) high flow pulses, (iv) small floods, and (v) high floods. The detailed information about the IHA parameters and EFCs are available in user manual (*https://www.conservationgateway.org/Documents/IHAV7.pdf*).

Since the flow in rivers are minimum during the drought period; therefore, this study considered on the ELF and LF components from the five major EFC classes to understand the impact of drought (i.e., threshold that triggers the ELF and LF) on EF.

Line 191) "widely used to assess hydrological alterations". Please include references.

**Response:** The following references are added to the manuscript to support the statement:

*Pal, S. and Sarda, R.: Measuring the degree of hydrological variability of riparian wetland using hydrological attributes integration (HAI) histogram comparison approach (HCA) and range of variability approach (RVA). Ecol. Ind., 120, p.106966, 2021.*

*Rahman, K.U., Shang, S., Shahid, M. and Wen, Y.: Hydrological evaluation of merged satellite precipitation datasets for streamflow simulation using SWAT: a case study of Potohar Plateau, Pakistan. J. Hydrol., 587, p.125040, 2020.*

*Shiau, J.T. and Wu, F.C., 2006. Compromise programming methodology for determining instream flow under multiobjective water allocation criteria 1. JAWRA J. Am. Water Res. Assoc., 42(5), pp.1179-1191.*

*Zheng, X., Yang, T., Cui, T., Xu, C., Zhou, X., Li, Z., Shi, P. and Qin, Y.: A revised range of variability approach considering the morphological alteration of hydrological indicators. Stoch. Environ. Res. Risk Assess., 35, pp.1783-1803, 2021.*

Line 197) "major steps in implementing RVA include..". However, the authors are not considering all those steps, which steps do the authors use and why?

**Response:** Authors have used most of the steps mentioned in the paragraph. For example, i) we have studied the hydrological alterations in streamflow using the rate and magnitude of flow in the rivers (ELF and LF), ii) we quantified the degree of alterations using the HAF (Hydrological Alteration Factor) and results are shown in Figures 7-8, iii) before the application of RVA analysis, our hypothesis tested in this study was that drought causes significant alterations in streamflow, iv) the above hypothesis was tested using threshold regression and HAF, and v) since we quantified the impact of drought (i.e., shown the terms of drought severity that causes ELF and LF (Table 2), time where the drought caused ELF and LF (Tables 3-5), and Figures 7-8), the results and recommendation in this study will help in implementing the necessary ecosystem measures across the Indus Basin.

Line 201-202) Why are the drought years considered a post-impact period and the whole period a pre-impact period? The impacts of drought are felt during drought and afterwards. Please explain this division.

**Response:** As mentioned in the guidelines of IHA and RVA, we have to divide the study period into pre-impact and post-impact periods to analyze the hydrological alterations. For instance, in order to analyze the impact of a structure (e.g., dam) built on river, we have to consider the entire study period as pre-impact (before the dam construction) and post-impact (after the dam

construction) period. However, decrease in the magnitude of streamflow might not only be associated with the drought. Therefore, we have considered the entire study period as a pre-impact period and the representative drought years as post-impact period.

Line 217-223) The authors explain here that threshold regression differs from change-point analysis. Why is this explained and why is threshold regression chosen instead of change-point analysis? Please elaborate on this better.

**Response:** The authors preferred the use of threshold regression over change-point analysis mainly on the bases of the following three reasons.

i) threshold regression is capable to understand the non-linear relationship between the threshold variables (drought and environmental flow in our case), while change-point analysis can be used to see the changing trend in a time-series data (for instance, we can only see the change point in drought or in environmental flow).

ii) threshold regression is more robust than change-point analysis in dealing with non-linear relationship between the variables, and comparable with other non-linear regression models (e.g., spline regression model).

iii) threshold regression has the potential to adapt any shape (explained by Fong et al., 2017) depending on the threshold variable and its threshold value

Line 225) How do the authors define drought severity in this study? Please explain this either in the Introduction or Methodology. For example, are specific gradations of SPEI used to consider drought severity (moderate drought, extreme drought etc.)?

**Response:** Drought is classified into different classes following the recommendations from McKee et al., (1993), where the classification is based on the Standardized Precipitation Index (SPI) algorithm.

1. **Results and Discussion**

Line 241) Where are the authors referring to with "Representative catchments"? Furthermore, this sentence is probably not necessary.

**Response:** The statement is removed as suggested.

Line 245) The authors mention here "extreme, severe, and moderate drought events", coming back to my earlier comment, how are those defined?

**Response:** An explanation to drought severity classes has been added in the methodology section.

Line 283-284) How are ELF, LF, high flow pulses, small floods, and large floods defined?

**Response:** As mentioned previously, the output from IHA is classified into IHA parameters (33) and EFCs (34). The EFC parameters are further divided into five major classes, including i) ELF, ii) LF, iii) high flow pulses, iv) small floods, and iv) large floods. The detailed information about

further division of these five main classes into other sub-classes can be found in the user manual (*https://www.conservationgateway.org/Documents/IHAV7.pdf*).

Table 2) Indicate in the table that the first values are related to ELF and the second to LF.

**Response:** The Table 2 actually shows the drought values that resulted in ELF and LF. Since we are correlating the ELF and LF to drought, we did not differentiate between the two EFC parameters in this study because both LF and ELF are mostly linked with drought (no precipitation for longer period thus turning meteorological drought into hydrological drought and results in LF and ELF). Therefore, the table only shows the drought severity (values) that causes ELF and LF in each of representative catchments in the Indus Basin.

Line 347) "three and four", is meant here "three locations"?

**Response:** The Table 3 shows the time zone for a specific threshold (i.e., drought severity) where there are no significant alterations in the flow as explained in lines 341-342. The "three and four" mentioned by authors are not associated with locations, it is actually associated with the time zones where the flow in rivers have no significant alterations. It can also bee seen in the table; for instance, the entire study period in Gilgit catchment is divided into three time zones (i.e., 1980-1991, 1992-2011, and 2012-2018).

Line 395) The 18 EFC components is coming a bit out of the blue, what are those 18 components?

**Response:** As explained previously, the five major EFC components are further divided into different sub-classes. Here in this study, we have picked 18 sub-classes from the ELF and LF main classes. These 18 sub-classes include the magnitude, frequency, and rate of ELF and LF at each month (12 sub-classes from January to December) and 6 classes (including 1-day, 7-day, 30-day, and 90-day minimum flows, low pulse count and low pulse duration.

Line 397) "Overall, environmental flow … catchments in LIB": why is this the case?

**Response:** The vulnerability of catchments to drought in terms of ELF and LF is categorized into no alterations, moderate alterations and high alterations using the HAF. The conclusion mentioned in lines 397-398 is based on the Figure 7 and explained in the text.

Figure 7) Please include more information in the caption; is it averaged over which time period, which years etc.?

**Response:** Corrected. It is averaged over the entire study period.

There is no discussion (either in the Introduction or Discussion) about drought literature in the Indus basin in general; did other authors use other methods to look at flows in the Indus basin in relation to drought and what makes this study so innovative? Please elaborate on this in the Introduction and/or Discussion (to compare it with the results of this study). In addition, it is probably interesting to address catchment memory in relation to drought as it plays a huge role in

drought impact on river flows and the prediction of hydrological drought, see for example Sutanto & Van Lanen (2022).

**Response:** The discussion section is added into the manuscript following the recommendation of both reviewers. Actually, this study is first of kind to link drought with environmental flow using threshold regression; therefore, very limited literature is available on this topic. However, the discussion section is added and novelty is highlighted following the guidelines and suggestions from reviewers.

Authors are thankful to reviewer for the suggestions related to linking the current study with catchment memory. Catchment memory, as explained by Sutanto and Lanen et al. (2022), plays an important role in the transition of meteorological drought to hydrological drought in the most upstream catchments without flow from upstream. However, for catchments in MIB and LIB, flows are mainly from the corresponding upstream UIB/MIB catchments and less from local runoff. Therefore, catchment memory is likely to have a minor influence on the hydrological drought in the MIB and LIB catchments. As the study catchments cover the whole Indus Basin in Pakistan, we did not add analysis related with catchment memory in this manuscript. However, analysis related with catchment memory for the UIB catchments can be carried out in our future study.

1. **Conclusion**

Line 456) "The analyses have … occurrence of droughts": how did the authors show that temperature plays an important role in the occurrence of drought? This study does not compare drought indicators or look specifically at evapotranspiration or other definitions of drought.

**Response:** Actually, the climate of catchment in the LIB are mostly hyper-arid in nature and this study found that drought is more severe in LIB than in other catchments of the Indus Basin. Therefore, this conclusion was made that temperature plays a critical role in the onset of drought. However, no such detailed analyses are carried out in this study and the statement is removed from the revised manuscript.

Line 465) "In other words" does not really seem appropriate here as the text is not summarizing what is said before.

**Response:** The term "In other words" has been removed in the revised manuscript.

Line 475-476) "In addition, the … in specific catchments": the SPEI coefficient increases with increasing drought; do the authors mean the accumulation periods? At the moment, this sentence is not clear.

**Response:** The statement is revised as "In addition, the SPEI coefficient from threshold regression in scenario-2 (shown in Tables 3 and 4) increases with increase in drought severity, suggesting that SPEI has a significant impact on environmental flow in specific catchments".

**References:**

Sutanto, S. J., & Van Lanen, H. A. (2022). Catchment memory explains hydrological drought forecast performance. Scientific Reports, 12(1), 2689.

Van Loon, A. F. (2015). Hydrological drought explained. Wiley Interdisciplinary Reviews: Water, 2(4), 359-392.

---

## Referee Report (RR1)

**'Catchment scale assessment of drought impact on environmental flow in the Indus Basin, Pakistan'
by Rahman et al.**

This study entails the critical role of understanding drought impacts on environmental flows and contributes valuable insights into preserving the ecological integrity of rivers in the Indus Basin. In this study, the authors use a combination of various methods for analyzing drought and environmental flows. Their aim to inform and facilitate sustainable water resources management practices is vital in the region. With the projected increase in the frequency of future extreme droughts in the Indus catchment, this study could add valuable insights in understanding environmental flows in the catchment and be a good addition to the scientific literature. However, less focus should be put on the drought events and drought severity in the catchments (because this is already know in literature) and more on the novelty of the study which is the influence of drought on the magnitude of the extreme low flows and low flows. I suggest some improvements detailed below, that may help in bringing this message out more clearly before the manuscript can be viable for publication. I hope my comments can contribute to enhancing the quality of the paper.

**Major comments**

- First of all, the question of uncertainty in the datasets, and the IHA model used should be discussed in the discussion section. How this would have impacted the analysis? I understand that the research idea is to understand how droughts influence extreme low flows and low flows, but perhaps more details on the different methods used could be given in the methodology, for example, how did you select the threshold for the threshold regression analysis, what does the coefficient value represent and also further elaboration on the steps involved in the implementation of the range variability analysis could be provided instead of a list. Generally, the methodology on range variability analysis and threshold regression analysis could be elaborated further to enable easier understanding on the methods i.e. methods for performing threshold regression analysis is a bit lacking hence when one reaches the results section, understanding becomes difficult.
- Secondly, the authors performed further data quality tests on the hydro-meteorological datasets i.e. using kurtosis and skewnesss, these plots could be included in the supplementary materials, this may help with the question on uncertainty of the datasets. For example, Line 144: The authors state that the data was thoroughly analyzed and the period from 1980-2018 chosen. How was this analysis carried out? Additionally, the authors failed to discuss the limitations of the research and future recommendations. How does this research compare to other studies done in the Indus catchment? Generally, the discussion of the results could be improved and line 496-506 could be moved to introduction.
- On a side note, the authors could try using threshold-based indices instead of the standardized indices. This is to explicitly bring out the roles of temperature and precipitation on the occurrence of droughts and then make the conclusion on influence of temperature on drought occurrence as done in line 516: 'The analyses have shown that temperature plays a crucial role in the occurrence of droughts'.
- Additionally, in the results section, it is hard to differentiate what is moderate and what is extreme drought because this is not indicated. What is considered a low flow and extreme low flow? Is it possible to provide specific values for these magnitude of the low and extreme flows associated with each of the drought severities?

- In general, the quality of the writing and preparation should be improved. I found it hard to continuously scroll up to always look for the full meaning of the abbreviations. It is good practice to state the full abbreviations again especially when it comes at the beginning of the paper. What I would suggest to the authors is that they carefully review the text to avoid several grammar errors and typographical errors prevalent in the manuscript
- Line 165: How did the authors calculate the potential evapotranspiration or actual evapotranspiration used in the water balance equation?
- Line 178: How did the authors come to this choice of threshold value?
- Line 215-line 216: This statement doesn't make sense. Do you mean the no drought years are pre-impact and drought years post-impact? How did you come to select SPEI-12 for the analysis? Isn't that double counting or rather counterintuitive cause the drought years are already in the pre-impact period, if the whole period was considered? How does this impact your results? Or did you remove the drought years? If yes, then state this instead of saying the whole period was considered as pre-impact period
- Line 221: The numbers are very specific, is this based on something? If that is the case please cite or if not give reasons for the categories division

**Minor comments**

- The authors should carefully review the text to avoid several grammar errors and typographical errors prevalent in the manuscript for example;
  - Line 38: 'On a global scale…' instead of 'At a global scale…'
  - Line 56: '…sustainable management of surface and groundwater…'
- In Figure 1, it would be better if the authors combine the legends so you have a single legend to indicate the basin names
- Line 260: I would suggest to mention the list of names based on the figure alignments
- Line 279: It would be clearer if the authors indicated these years on the plots, otherwise it becomes difficult to look for the specific years within the each of the plots. I would suggest to do the same for figures 3-5
- Figure 4: Use the same scale for all the plots. It becomes quite confusing when they have different scales for example first look makes me think that Hunza catchment droughts are more severe. I would suggest to do the same for the rest of the plots (figures 3-5, 6)
- Line 294: I think this statement should be moved to the case study section
- Line 364: Jhelum Rive is also divided into three time periods
- Line 358: As a reader, I find using the term time zones confusing, is there a better term that could be used instead? e.g. time period?
- Table 2: Is it possible for the authors to Separate the catchments to UIB, MIB and LIB, otherwise one keeps going up to the case study section to check which catchments are where. Additionally, is it possible to indicate the specific thresholds for extreme low flows and low flows instead of combination?

---

## Author Response (AR2)

**Response to Reviewer Comments**

This study entails the critical role of understanding drought impacts on environmental flows and contributes valuable insights into preserving the ecological integrity of rivers in the Indus Basin. In this study, the authors use a combination of various methods for analyzing drought and environmental flows. Their aim to inform and facilitate sustainable water resources management practices is vital in the region. With the projected increase in the frequency of future extreme droughts in the Indus catchment, this study could add valuable insights in understanding environmental flows in the catchment and be a good addition to the scientific literature. However, less focus should be put on the drought events and drought severity in the catchments (because this is already known in literature) and more on the novelty of the study which is the influence of drought on the magnitude of the extreme low flows and low flows. I suggest some improvements detailed below, that may help in bringing this message out more clearly before the manuscript can be viable for publication. I hope my comments can contribute to enhancing the quality of the paper.

**Response:** Authors are thankful to the valuable comments from reviewer that helped in substantially improving the quality of the manuscript. The comments are incorporated into the manuscript (can be seen through track changes) and response of each comment is given here in blue color.

**Major Comments:**

1.  First of all, the question of uncertainty in the datasets, and the IHA model used should be discussed in the discussion section. How this would have impacted the analysis? I understand that the research idea is to understand how droughts influence extreme low flows and low flows, but perhaps more details on the different methods used could be given in the methodology, for example, how did you select the threshold for the threshold regression analysis, what does the coefficient value represent and also further elaboration on the steps involved in the implementation of the range variability analysis could be provided instead of a list. Generally, the methodology on range variability analysis and threshold regression analysis could be elaborated further to enable easier understanding on the methods, i.e., methods for performing threshold regression analysis is a bit lacking hence when one reaches the results section, understanding becomes difficult.

**Response:** The analysis to understand uncertainty in collected data including in-situ precipitation and temperature data are already carried out by the authors in previously published manuscripts, and also the descriptive statistics of the data is given in the supplementary file to this manuscript. Data processing before carrying out the analysis is described in Lines 156-161 (in the revised manuscript, the same below). Some of the insights into data quality and IHA analyses are further discussed in the discussion section.

This research is mainly devoted to understand the impact of drought on environmental flow, where threshold regression is a robust technique to understand the impact of drought on environmental flow as threshold regression not only identifies the drought severity but also the time when drought has resulted in significant decline of flow in major rivers of the Indus Basin. The advantages and effectiveness of threshold regression over other change-point analysis method are already discussed in details in Lines 248-258.

We add more explanation on the results of threshold regression analysis: The coefficient in Table 3 quantifies the impact of drought on environmental flow. For instance, the threshold regression in Gilgit catchment shows that drought has a significant impact (0.949) on environmental flow during the period of 1992-2011. During the mentioned period, Indus Basin has experienced frequent extreme drought events which not only impacted the surface water availability but also other sectors including agriculture (Rahman et al., 2023). It should be noted that the coefficient of SPEI-1 varies significantly from one catchment to another and from one period to another during to significant variations in climatic and land use characteristics accompanied with frequent fluctuations in SPEI-1 estimates (Lines 379-384).

The threshold regression is run under two different scenarios to understand what is the magnitude of drought that causes the extreme low flows and low flows, and when (time) the drought resulted in extreme low flows and low flows. Explanation to these points is already given in lines 333-337.

2. Secondly, the authors performed further data quality tests on the hydro-meteorological datasets, i.e. using kurtosis and skewnesss, these plots could be included in the supplementary materials, this may help with the question on uncertainty of the datasets. For example, Line 144: The authors state that the data was thoroughly analyzed and the period from 1980-2018 chosen. How was this analysis carried out? Additionally, the authors failed to discuss the limitations of the research and future recommendations. How does this research compare to other studies done in

the Indus catchment? Generally, the discussion of the results could be improved and line 496-506 could be moved to introduction.

**Response:** The descriptive statistics of the kurtosis and skewness results are given in a Tables S1 and S2, which clearly depicts that data is normally distributed. The data is then used to calculate the drought indices SPEI.

Explanation is added to line 144: After thoroughly analyzing all the collected data (i.e., checking the date/years of available data at most of the in-situ stations), a period from 1980–2018 is chosen to demonstrate the drought impact on environmental flow where all the in-situ stations have the data with few or no missing values (Lines 150-152).

Limitations of the research and future recommendations are added in the discussion section (Lines 522-533).

The original Lines 500-505 were moved to the introduction section (Lines 75-79).

3. On a side note, the authors could try using threshold-based indices instead of the standardized indices. This is to explicitly bring out the roles of temperature and precipitation on the occurrence of droughts and then make the conclusion on influence of temperature on drought occurrence as done in line 516: 'The analyses have shown that temperature plays a crucial role in the occurrence of droughts'.

**Response:** The authors have used the most commonly used standardized drought index (i.e., SPEI) to monitor droughts' impact on environmental flow. The use of SPEI for this purpose is of significant importance as several studies have estimated drought using SPEI. Further, the National Disaster Management Authority (NDMA) of Pakistan recommended SPI and SPEI to monitor drought in Pakistan. Therefore, this study has used SPEI to analyze the impact of drought on environmental flow, which can be easily interpreted and help policy makers to devise a policy that mitigates the impact of drought on environmental flow. Authors have published several manuscripts that highlighted the role of temperature on the occurrence of drought events over the Indus Basin. Moreover, this study also highlighted that as we move from UIB to LIB, the frequency and severity of drought significantly increase which substantially reduced the environmental flow.

**References:**

Ur Rahman, K., Shang, S., Balkhair, K. and Nusrat, A., 2023. Catchment-Scale Drought Propagation Assessment in the Indus Basin of Pakistan Using a Combined Approach of Principal Components and Wavelet Analyses. Journal of Hydrometeorology, 24(4), pp.601-624.

Rahman, K.U., Hussain, A., Ejaz, N., Shang, S., Balkhair, K.S., Khan, K.U.J., Khan, M.A. and Rehman, N.U., 2023. Analysis of production and economic losses of cash crops under variable drought: A case study from Punjab province of Pakistan. International Journal of Disaster Risk Reduction, 85, p.103507.

Hussain, A., Jadoon, K.Z., Rahman, K.U., Shang, S., Shahid, M., Ejaz, N. and Khan, H., 2023. Analyzing the impact of drought on agriculture: evidence from Pakistan using standardized precipitation evapotranspiration index. Natural Hazards, 115(1), pp.389-408.

4. Additionally, in the results section, it is hard to differentiate what is moderate and what is extreme drought because this is not indicated. What is considered a low flow and extreme low flow? Is it possible to provide specific values for these magnitude of the low and extreme flows associated with each of the drought severities?

**Response:** Drought severity is divided into several categories (a standardized procedure followed in several manuscripts) based on the SPEI values. In this revised version, we have added the SPEI ranges of drought severity classes in Lines 186-187 and Table S3.

The flow is considered low flow when the magnitude of flow is less than $25^{th}$ percentile (Kumar et al., 2022). In this study, when the magnitude of flow is less than $10^{th}$ percentile, we classified it as extreme low flow. The division of flows into different classes and their ranges are given in Lines 200-203.

"IHA categorizes streamflow into several components, including low flows (where the streamflow values are less than or equal to $25^{th}$ percentile), moderate flows (where the streamflow values range between $26^{th}$ to $75^{th}$ percentile) and high flows (where the streamflow values are greater than $75^{th}$ percentile). Besides, when the flow is less than the 10th percentile, we classified it as extreme low flow".

5. In general, the quality of the writing and preparation should be improved. I found it hard to continuously scroll up to always look for the full meaning of the abbreviations. It is good

practice to state the full abbreviations again especially when it comes at the beginning of the paper. What I would suggest to the authors is that they carefully review the text to avoid several grammar errors and typographical errors prevalent in the manuscript.

**Response:** The abbreviations are checked and listed at the end of the manuscript. Moreover, the manuscript is carefully checked and revised to remove grammatical and typographical errors.

6. Line 165: How did the authors calculate the potential evapotranspiration or actual evapotranspiration used in the water balance equation?

**Response:** SPEI calculation is usually based on potential evapotranspiration, which can be calculated using several methods, including Thornthwaite, Penman-Monteith, and Hargreaves. However, we have used Hargreaves equation in this method to calculate potential evapotranspiration as it requires less data (Hargreaves and Samani, 1985). The remaining methods are based on extensive data including solar radiation, relative humidity, and wind speed, which is mostly not available in the Indus Basin. Therefore, Hargreaves method is suggested by several authors when other data is not available (Abbasi et al., 2021; Allen et al., 1996; Dubrovsky et al., 2009).

**References:**

*Hargreaves, G.H. and Z.A. Samani, Reference crop evapotranspiration from temperature. Applied engineering in agriculture, 1985. 1(2): p. 96-99.*

*Dubrovsky, M., M.D. Svoboda, M. Trnka, M.J. Hayes, D.A. Wilhite, Z. Zalud, and P. Hlavinka, Application of relative drought indices in assessing climate-change impacts on drought conditions in Czechia. Theoretical and Applied Climatology, 2009. 96: p. 155-171*

*Abbasi, A., K. Khalili, J. Behmanesh, and A. Shirzad, Estimation of ARIMA model parameters for drought prediction using the genetic algorithm. Arabian Journal of Geosciences, 2021. 14(10): p. 841.*

*Allen, R.G., Assessing integrity of weather data for reference evapotranspiration estimation. Journal of irrigation and drainage engineering, 1996. 122(2): p. 97-106.*

7. Line 178: How did the authors come to this choice of threshold value?

**Response:** Drought severity, as mentioned above, is divided into several classes, including normal (wet/drought events) with SPEI values ranging between 1 to -1. To avoid the frequent normal events, we have considered a threshold of SPEI < -1 (Table S3).

8.  Line 215-line 216: This statement doesn't make sense. Do you mean the no drought years are pre-impact and drought years post-impact? How did you come to select SPEI-12 for the analysis? Isn't that double counting or rather counterintuitive cause the drought years are already in the pre-impact period, if the whole period was considered? How does this impact your results? Or did you remove the drought years? If yes, then state this instead of saying the whole period was considered as pre-impact period.

**Response:** RVA is typically used to understand the post-impact of any alteration in streamflow, where the entire period is divided into pre- and post-impact periods on the representing the streamflow before and after the alteration, respectively. In the current study, we considered the impact of drought by comparing the entire period (1980–2018) with specific drought years (i.e., years/representative drought years where the SPEI values are less than -1) to understand how drought events causes alterations in streamflow. The statement about SPEI-12 is revised as "To investigate the impact of drought on environmental flow in the current study, the whole period (1980-2018) is considered a pre-impact period (without differentiating between drought and wet events, which can also be considered as normal flow years without focusing on specific drought years), while the specific drought years (i.e., where the average SPEI values are less than -1, also considered as representative drought years as identified by Rahman et al., 2023a and 2023bSPEI-12 < -1) are considered a post-impact period" given in lines 228–232.

In the whole period, we have considered the streamflow from 1980–2018 without differentiating between streamflow during drought and wet years. In other words, the pre-impact period is equivalent to the streamflow without differentiating between drought or wet periods. In contrast, if we consider the years excluding drought years, it will not meet the conditions of pre-impact period as mentioned in the RVA guidelines.

9.  Line 221: The numbers are very specific, is this based on something? If that is the case please cite or if not give reasons for the categories division

**Response:** HAF values are already defined in the literature by Ritcher et al. (1997). Citation is added to the lines as suggested (Line 235).

**Minor Comments:**

1. The authors should carefully review the text to avoid several grammar errors and typographical errors prevalent in the manuscript for example;

**Response:** The manuscript is checked for grammatical errors and typos as suggested.

2. In Figure 1, it would be better if the authors combine the legends so you have a single legend to indicate the basin names.

**Response:** The Figure 1 shows four different aspects of the Indus Basin; (a) elevation, (b and c) the location of rainfall/temperature and flow gauges along with the demarcation of Upper, Middle, and Lower Indus Basins, and (d) sub-division of the three basins into different catchments. For better representation of the catchments, the number along with specific names of the catchment is given separately. Therefore, it is better to represent different aspects of the Indus Basin separately.

3. Line 260: I would suggest to mention the list of names based on the figure alignments.

**Response:** Corrected as suggested.

4. Line 279: It would be clearer if the authors indicated these years on the plots, otherwise it becomes difficult to look for the specific years within the each of the plots. I would suggest to do the same for figures 3-5.

**Response:** Corrected as suggested.

5. Figure 4: Use the same scale for all the plots. It becomes quite confusing when they have different scales for example first look makes me think that Hunza catchment droughts are more severe. I would suggest to do the same for the rest of the plots (figures 3-5, 6).

Response: Corrected.

6. Line 294: I think this statement should be moved to the case study section.

Response: We think this statement is a continuation of previous statements and do not move it to the case study section.

7.  Line 364: Jhelum Rive is also divided into three time periods

**Response:** Corrected.

8.  Line 358: As a reader, I find using the term time zones confusing, is there a better term that could be used instead? e.g. time period?

**Response:** Time zones are replaced with time periods as suggested.

9.  Table 2: Is it possible for the authors to Separate the catchments to UIB, MIB and LIB, otherwise one keeps going up to the case study section to check which catchments are where. Additionally, is it possible to indicate the specific thresholds for extreme low flows and low flows instead of combination?

**Response:** Table 2 is revised as suggested; however, it is not possible to specify separate thresholds for extreme low flows and low flows as threshold regression separates/identifies on time period from another on the basis of a specific threshold (i.e., sudden shock or break in the time series) irrespective of the magnitude of streamflow provided. We will consider this problem in further studies.